# Transcriptional and morphological profiling of parvalbumin interneuron subpopulations in the mouse hippocampus

Lin Que[1,2], David Lukacsovich[1,2], Wenshu Luo[1] & Csaba Földy [1✉]

The diversity reflected by >100 different neural cell types fundamentally contributes to brain function and a central idea is that neuronal identity can be inferred from genetic information. Recent large-scale transcriptomic assays seem to confirm this hypothesis, but a lack of morphological information has limited the identification of several known cell types. In this study, we used single-cell RNA-seq in morphologically identified parvalbumin interneurons (PV-INs), and studied their transcriptomic states in the morphological, physiological, and developmental domains. Overall, we find high transcriptomic similarity among PV-INs, with few genes showing divergent expression between morphologically different types. Furthermore, PV-INs show a uniform synaptic cell adhesion molecule (CAM) profile, suggesting that CAM expression in mature PV cells does not reflect wiring specificity after development. Together, our results suggest that while PV-INs differ in anatomy and in vivo activity, their continuous transcriptomic and homogenous biophysical landscapes are not predictive of these distinct identities.

[1] Laboratory of Neural Connectivity, Brain Research Institute, Faculties of Medicine and Science, University of Zürich, Zürich, Switzerland. [2]These authors contributed equally: Lin Que, David Lukacsovich. ✉email: foldy@hifo.uzh.ch

A central goal in brain research is the clear and comprehensive classification of cell types according to anatomical and physiological features, as well as by distinct molecular markers that unambiguously identify each type[1–7]. Recent single-cell RNA-seq assays have facilitated classification efforts[8–12] and increased expectations that transcriptomic information could explain the entirety of cell-type-specific features, a crucial step toward understanding the multimodal identity of neural cell types. While several studies have employed single-cell RNA-seq to characterize transcriptomic and physiological features[12–21], the relationship between transcriptomic information and neuronal morphology is the subject of on-going research[14,20–22].

PV-INs of the CA1 hippocampus are a particularly suitable model to study this problem, as they appear to be a biophysically homogenous population[23], yet have distinct morphological types[4]. Morphologically, CA1 PV-INs can be divided into three main cell classes based on their axonal projections: (1) axo-axonic cells (AAC) that specifically project to the axon initial segment of pyramidal cells; (2) basket cells (BC), which establish synapses onto the perisomatic region of the postsynaptic neuron and thus restrict their axons to the pyramidal cell layer; (3) and bistratified cells (BIC), whose axons target more distal dendrites and thereby extend their axons specifically in the oriens and radiatum[7,24]. Based on dendritic morphology, each PV class can be further subdivided into horizontal (h; dendrites remain in stratum oriens) and vertical (v; dendrites extend along the stratum oriens-radiatum axis) types (hAAC, vAAC, hBC, vBC, hBIC, vBIC, respectively (Supplementary Fig. S1). The individual PV types display different activity patterns in vivo. During theta rhythms, successive activation of AAC (at the theta peak), BC (at descending phase), and BIC (at trough) types provide chronologically organized inhibition of pyramidal cells along their axon, soma, and dendrites[4]. Overlaid onto this, during fast network events, horizontal and vertical PV-INs display higher and lower activity, respectively, within each of the AAC, BC, and BIC classes[25].

Contrasted with the anatomical and functional differences, a seminal single-cell RNA-seq study on hippocampal CA1 interneurons found that transcriptomically, PV-INs comprise a largely continuous population, divided into two transcriptomic types of approximately equal prevalence, *Pvalb.Tac1* (268 cells) and *Pvalb.C1ql1* (211 cells; Harris et al.[10]; GSE99888; hereafter we refer to this as the "CA1-IN study"). In absence of morphological information, this study suggested that these populations represent the AAC and combined BC/BIC types, respectively. Consistent with this transcriptomic continuity, a certain continuity between morphological PV types is also presumed to exist, where cells may display overlap of multiple morphological characteristics of different canonical types[7,26].

A separate central hypothesis postulates that synaptic cell surface recognition and adhesion molecules (commonly referred to as CAMs, for short) determine neuronal connectivity and identity[27–32] and therefore suggests that CAM content of the distinct morphological PV types, which display distinct wiring, should also be different. Consistent with this hypothesis, single-cell transcriptomics revealed specific CAM expression in multiple neuronal types[9,11,13,19,33,34]. Furthermore, our research revealed significant CAM differences between cell types of different developmental origins[35], including the MGE-derived PV and CGE-derived CCK interneurons in CA1[13]. However, these studies have also highlighted a higher than expected CAM similarity among cell types with the same developmental origin, without regard to their path of maturation or final wiring properties in the adult brain.

During circuit maturation, hippocampal GABAergic inhibition shows dynamic changes[36–42], and parvalbumin protein levels continue to increase[43]. In fast-spiking interneurons, extensive synaptic maturation has been described to take place in the first four postnatal weeks[38,44], which are correlated with transcriptional regulation of thousands of genes[45]. Furthermore, Er81[46] and ErbB4[47] levels were found to correlate with activity and plasticity, and whose expression may define cell states that arise along the cells' developmental trajectory[34,48,49] or as a consequence of neuronal activity[11].

To investigate the relation between transcriptomic and morphological cell identities, we collected single-cell RNA-seq data from morphologically identified PV-INs in the CA1 and tested the following three hypotheses: (1) transcriptomic profiles correspond to morphological cell types, (2) CAM diversity within a single neuronal family, such as PV-INs, could account for distinct connectivity, and (3) transcriptional profiles correlate with the postnatal developmental trajectories of hippocampal PV-INs. Using differential gene expression analysis, our results show highly similar transcriptomic profiles across the whole PV population and CAM expression that is independent of the cells' morphological identity and connectivity. Finally, transcriptomic changes in PV-INs were assessed over the course of circuit maturation, which reveals a sharp transcriptomic transition 3 weeks after birth that includes a rapid and stable onset of hemoglobin gene expression.

## Results

**Morpho-transcriptomic characterization of PV-INs.** To generate electrophysiological, morphological, and transcriptional data from PV-INs, we performed patch-clamp recordings on tdTomato+ cells in the CA1 region of the hippocampus, in brain slices prepared from PV-Cre::Ai14 mice. During recordings, cells were stained with biocytin, which allowed for post hoc morphological analysis, and after patch-clamp recordings, the cytosol was aspirated for subsequent RNA-seq[13] (Fig. 1a). After morphological analysis, cells were characterized as vertical or horizontal AAC, BC, or BIC type (Fig. 1a and Supplementary Fig. S1), to which we will refer to as "morphological PV types" for short. From mice that were at least 21-day old, we recorded 224 cells, of which 67 cells were classified as either morphological PV types and passed bioinformatic quality control (7 vAAC, 9 vBIC, 11 hBIC, 31 vBC, and 9 hBC; see "Methods"). Note that none of the recorded cells resembled hAAC morphology[50] and thus we could not include this type in our further analysis. To date, only six hAACs have been documented (four in rat[50] and two in mice[25]), underscoring that this type might be rarely detectable[24]. To complement our data, we included our SST-OLM dataset adapted from Winterer et al.[18] as a control MGE-derived, but non-PV hippocampal interneuron type (Supplementary Fig. S1b).

Transcriptomic analysis showed consistent expression of GABAergic and absence of glutamatergic markers in both PV and SST-OLM cells and expression profiles of specific interneuron markers *Pvalb* and *Sst* were consistent with cell-type identity (Supplementary Fig. S2). Using edgeR[51], 168 genes (out of 8586 that passed preliminary feature selection before we ran edgeR; see "Methods") were found to be enriched in PV compared to SST cells, and 93 genes were highly expressed in SST compared to PV-INs with FDR (*P*-adjusted) <0.05 and fold change >2 (top ten genes of each comparison are shown in Fig. 1c and Supplementary Fig. S2). Consistent with previous studies[10,11,34,52], *Erbb4* and *Tac1* were enriched in PV and *Npas1* in SST cells. Using UMAP (uniform manifold approximation and projection[53]), we performed dimension reduction on the transcriptomic data, which revealed that PV and SST cells separated into different groups, but morphological PV types did not distinctly cluster, even when plotted without SST cells

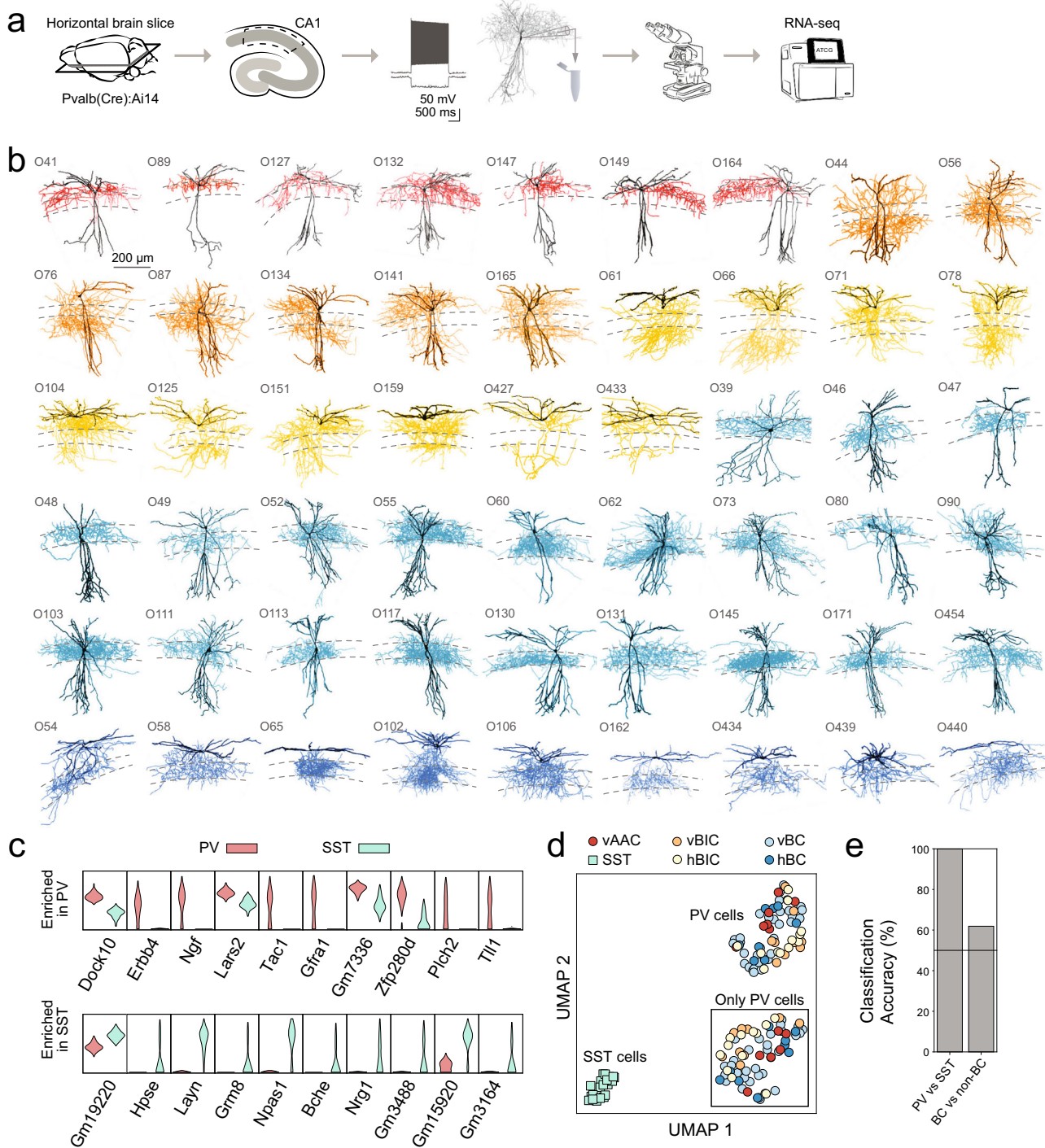

**Fig. 1 Morpho-transcriptomic profiling of PV-INs. a** Experimental pipeline: tdTomato+ PV-INs were recorded in acute brain slices, analyzed for their morphological and electrophysiological properties, and processed for single-cell RNA-seq. **b** Morphological reconstructions show different morphological PV types (red: vertical axo-axonic cell, vAAC, orange: vertical bistratified cell, vBIC, yellow: horizontal bistratified cell, hBIC, blue: vertical basket cell, vBC, dark blue: horizontal basket cell, hBC; these color code applies to all figures). **c** Violin plots show expression of top ten genes that are enriched in PV versus SST-OLM (upper plots) and in SST-OLM versus PV cells (lower plots). **d** Plot shows transcriptomic data-based dimension reduction using UMAP. Symbols correspond to different PV and the SST-OLM types. Insert shows UMAP for only PV-INs. **e** Bar plot shows classification accuracy for random-forest-based PV versus SST-OLM, BIC versus non-BIC, and for BC versus non-BC discriminations.

(Fig. 1d; also see Supplementary Fig. S3). By comparison, a supervised (random forest) classification algorithm accurately classified cells as PV or SST type (99.9% accuracy) but proved to be less efficient at (78.4% accuracy in case of the BIC versus non-BIC comparison) or insufficient (61.4% accuracy in case of the BC versus non-BC comparison, where sample numbers were sufficient to define separate training and testing sets) for distinguishing morphological PV types (Fig. 1e).

To further analyze any transcriptional differences in morphological PV types, we applied proMMT (probabilistic mixture modeling for transcriptomics[10]; a combined workflow for gene selection-based transcriptomic clustering) that was previously

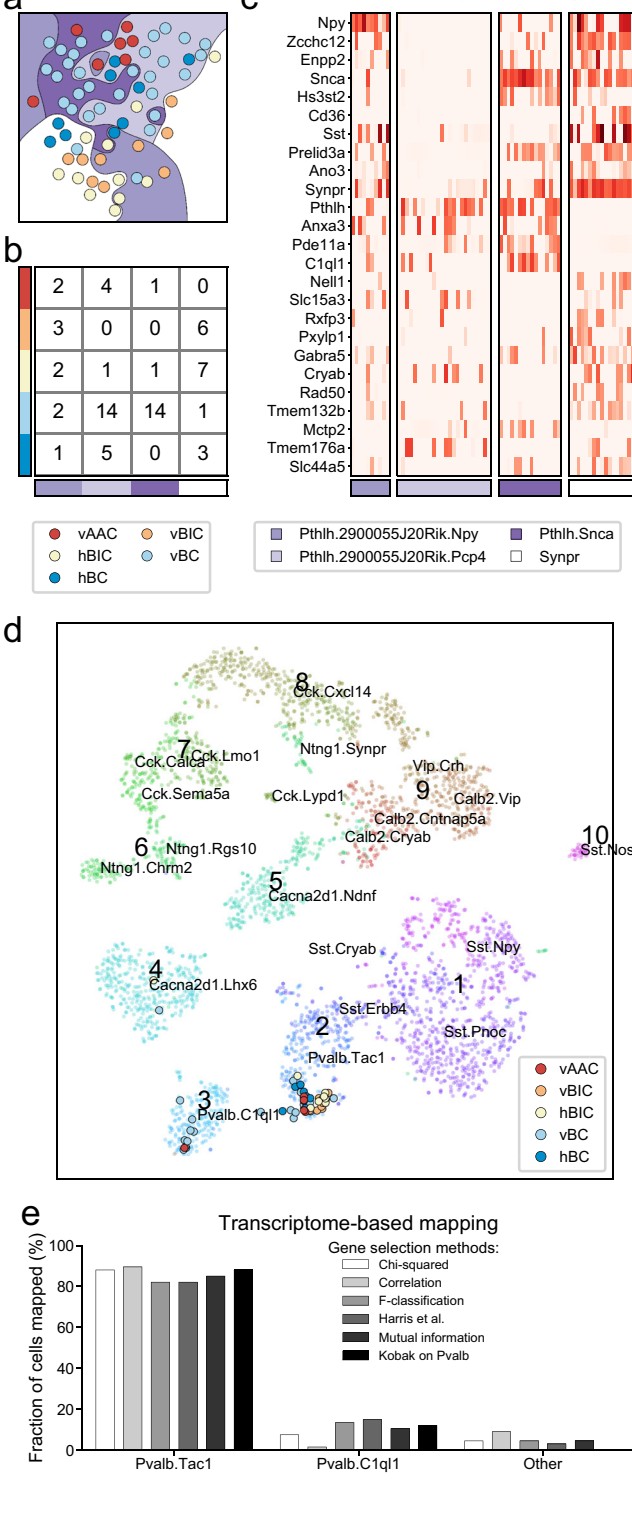

**Fig. 2 Transcriptomic properties of PV-INs. a** Plot shows unsupervised, transcriptomic data-based dimension reduction using nbt-SNE[10]. Single cells are labeled with circles, which are colored according to morphological classification. Background colors refer to transcriptomic classification using proMMT[10]. **b** Confusion matrix between proMMT-based transcriptomic types and morphological types. Numbers represent how many cells belong to both a given morphological type (rows) and a given transcriptomic type (columns). **c** Heatmap of all genes that were differentially expressed (FDR < 0.05 and fold change>2), in at least one pairwise comparison of transcriptomic types using edgeR. Cells are grouped by transcriptomic types. **d** Plot shows the mapping of PV-INs of this study onto the CA1-IN dataset[10]. Cell-type labels, e.g., *Pvalb.Tac1*, are also imported from the original CA1-IN study. Gene selection for mapping was performed using the method described in Kobak et al.[79]. **e** Quantification of mapping efficacy using six different gene selection methods (see "Methods" for details).

cell distributions within the proMMT types (Fig. 2b), which—together with the 78.4% classification efficiency shown in Fig. 1e—highlighted potential transcriptomic differences between BIC versus non-BIC types. To gain insight at the single-gene level, we plotted all differently expressed genes (FDR < 0.05, fold change >2) between any of the four proMMT types (Fig. 2c). In addition to *Synpr* and *Pthlh*, among others, this analysis revealed differential expression of *Npy*, *Sst*, *Zcchc12*, *Pde11a*, and *C1ql1* between the four proMMT types. To assess the transcriptional signatures of PV-INs in a larger context, we included the above referenced CA1-IN data. Meta-analysis of these data revealed that our cells showed unconstrained mapping onto the two cell populations that were identified as PV cells in the original study. Using six different gene selection methods, our PV-INs could be consistently mapped onto either of the two PV populations (Fig. 2d, e). However, morphological types did not correlate with the *Pvalb.Tac1* (presumed AAC) or *Pvalb.C1ql1* types (presumed BC/BIC population), suggesting that the clustering in the CA1-IN study did not arise from the three main PV types. In conclusion, these findings indicated that while morphological PV types are not majorly distinct transcriptionally from one another, key differences may exist between BIC versus non-BIC types.

**PV-INs comprise a biophysically homogenous population.** To assess whether yet un-covered biophysical differences further characterized morphological PV types, we quantified ten electrophysiological parameters, including passive (e.g., input resistance and membrane capacitance) and active (e.g., properties of single and train AP firing) membrane properties. However, pairwise comparisons for each electrophysiological property between the five morphological PV types (total of 100 comparisons) did not reveal statistically significant differences (Fig. 3a). To further corroborate, we used UMAP for dimension reduction to assess potential clustering among the electrophysiological parameters. Such clusters may arise due to a combination of electrophysiological differences, which are not necessarily significant, but together differentiate morphological or transcriptomic types. We considered the following biologically relevant scenarios: two dendro-morphological types (vertical versus horizontal distinction), three axo-morphological types (AAC, BC, and BIC), four proMMT transcriptomic types (as in Fig. 2a), and the five morphological PV types. However, cells did not cluster along any of these distinctions (Fig. 3b), and clustering between any two of the electrophysiological parameters were also lacking (Supplementary Fig. S5). To conclude, these results confirm that variability in biophysical parameter space of PV-INs does not form discrete sub-clusters, neither correlates with morphological or transcriptomic modalities.

used to analyze CA1-IN data. ProMMT yielded four transcriptional clusters, which we visualized using nbt-SNE (negative binomial t-SNE[10]; Fig. 2a). Notably, these four transcriptional groups could not be split into distinct clusters in two-dimensional space, even when other dimension reduction techniques were applied (PCA, t-SNE, Flt-SNE, and UMAP; Supplementary Fig. S4). However, nbt-SNE highlighted a gradient between the morphological PV types. In addition, AAC and BC types frequently appeared in the *Pthlh*, whereas BIC in the *Synpr* proMMT types. Using a confusion matrix[14], we confirmed the

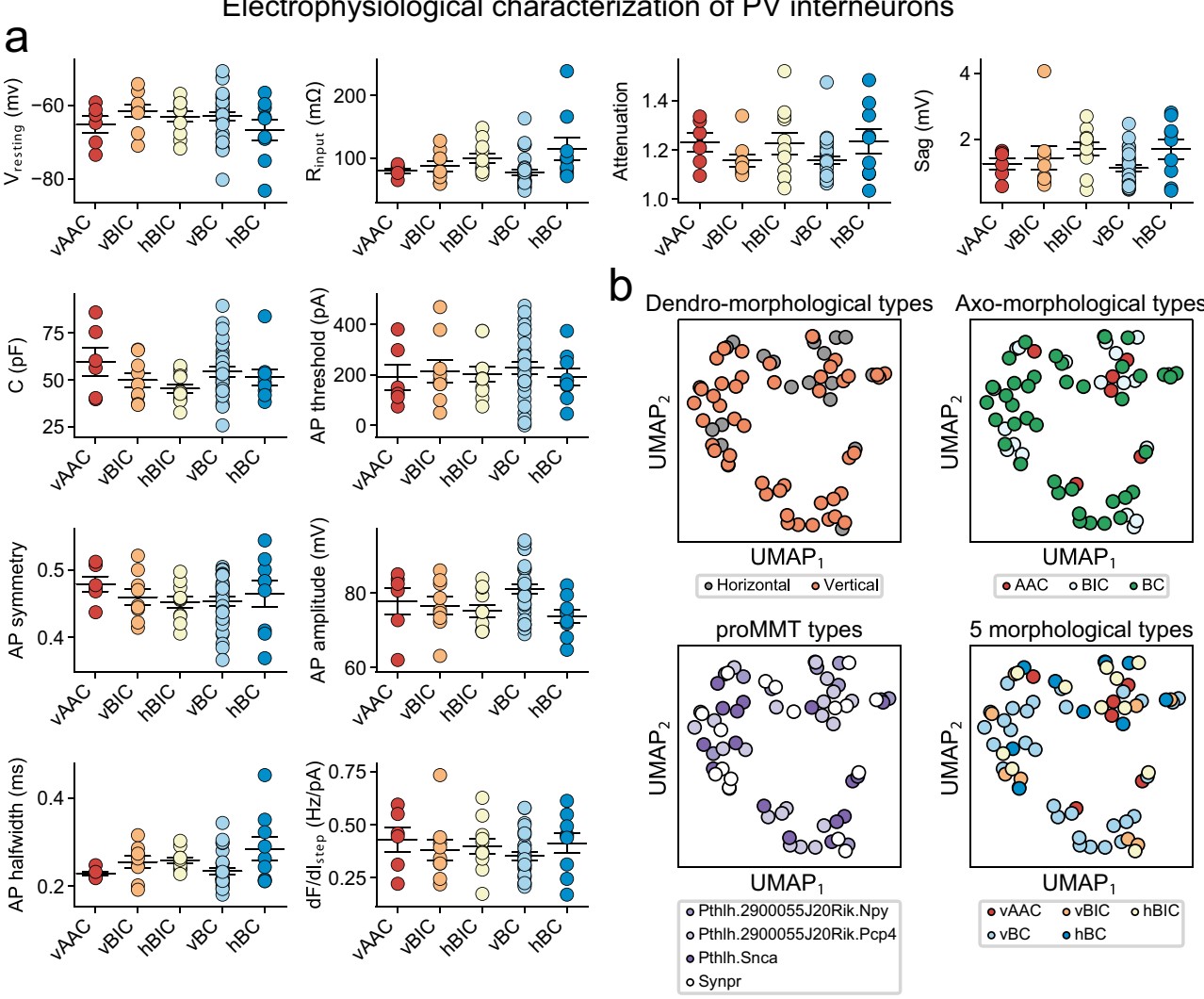

**Fig. 3 Electrophysiological properties of PV-INs. a** Plots show electrophysiological properties measured in the five morphological PV types. Circles represent single cells. Data represent mean ± S.E.M. **b** All panels show dimension reduction on electrophysiological data using UMAP. Circles represent single cells and are colored to show the two dendro-morphological (horizontal and vertical), three axo-morphological (AAC, BIC, and BC), four proMMT transcriptomic (Pthlh.2900055J20Rik.Npy, Pthlh.2900055J20Rik.Pcp4, Pthlh.Snca, and Synpr), and five morphological (vAAC, vBIC, hBIC, vBC, and hBC) PV types.

**Transcriptomic correlates of morphological PV types**. To study the correspondence between transcriptomic proMMT and morphological PV types, we next examined gene expression differences among the five morphological types, as well as among the axo- and dendro-morphological types. The 9 differentially expressed genes among the 5 morphological types (criteria were at least twofold difference in expression level and FDR < 0.05 between any two types using quasi-likelihood test) included *Akr1c18, Synpr, Pthlh* (all three in vAAC versus hBIC as well as vAAC versus vBIC comparisons), *Pcdh17, Zcch12, Sxbp6* (all three in vAAC vs vBIC comparisons), *Sst* (hBIC versus hBC and hBIC versus vAAC comparisons), *Npy* (vBIC versus hBC and vBIC versus vBC comparisons) and *Kcng4* (vAAC versus vBIC and vBIC versus vBC comparisons; Fig. 4a). PCA on these genes revealed a graded distribution of morphological PV types, from AAC to BIC type (Fig. 4a, lower panel). Comparison of axo-morphological types revealed already detected genes that distinguished BICs by enrichment (i.e., *Synpr, Cd36, Npy, Pthlh, Stxbp6,* and *Zcchc12*; see Fig. 2c), but also novel genes either lacking (e.g., *Gpc6, Kif16b, C1ql1*) or expressed (*Akr1c18*) in BIC types (Fig. 4b). Finally, the comparison of dendro-morphological

types revealed, among others, enrichment of *Luzp2, Ncam2, Fzd1,* and *C1ql1* in vertical and *Heatr5a* and *Zfx* in horizontal types (Fig. 4c; genes with FDR < 0.05 are shown). Using an extended set of 52 differentially expressed (FDR < 0.15; see Supplementary Information for complete list) genes from all three morphology-based comparisons, PCA revealed a continuity between the morphological PV types with a clearer separation of the morphological BIC and AAC/BC types (Fig. 4d), and the distinction between the hBIC and vBIC types (for implementation of support vector machine classification on this same problem and its conclusions, see Supplementary Fig. S6). In conclusion, morphology-based, supervised transcriptomic analyses revealed discrete patterns that identify BIC types, but did not further differentiate AAC and BC types. These results underscore the unsupervised feature selection and dimension reduction-based findings (Fig. 2) and further suggest that specific gene expression differences exist between BIC versus non-BIC types. However, gene expression in PV cells is largely similar and the whole population is characterized by a pronounced transcriptomic continuity, rather than by separate transcriptomic entities that correspond to each morphological type.

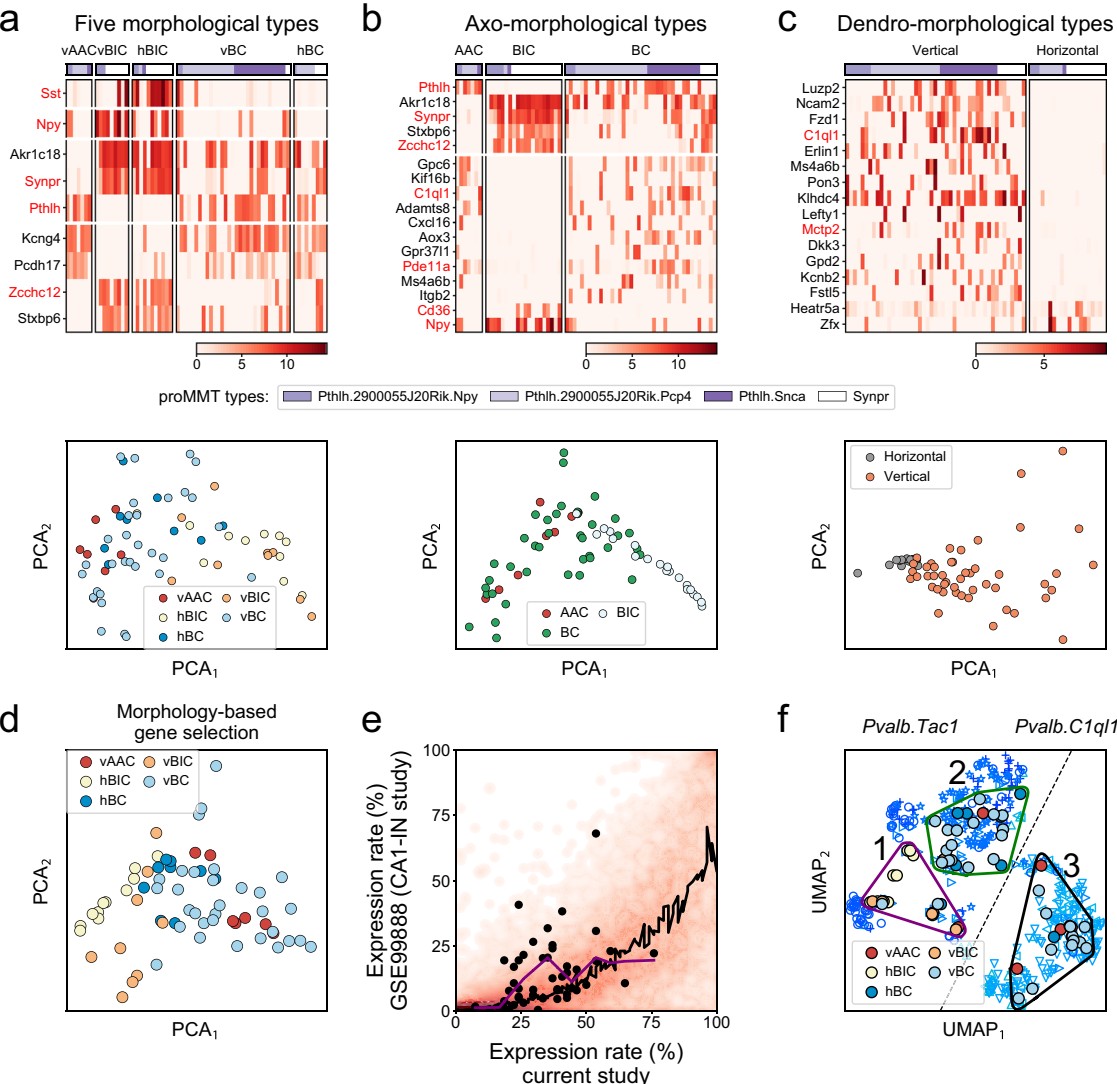

**Fig. 4 Transcriptomic correlates of morphological PV types. a–c** Heat maps (upper plots) of differentially expressed genes (using edgeR, fold difference>2, FDR < 0.05) between the five morphological PV types (panel **a**), three axo-morphological types (panel **b**), and two dendro-morphological types (panel **c**). Cells are ordered according to proMMT types. Genes that also appeared in proMMT comparisons (Fig. 2c) are highlighted in red. PCA plots (lower plots) were made using the differentially expressed genes. **d** PCA plot of cells, colored by the five morphological PV types, using an extended set (with a cutoff of FDR < 0.15) of $n = 52$ differentially expressed genes from panels **a** to **c**. **e** Comparison of expression rate of genes in PV-INs from this current versus the CA1-IN study[10], displayed as a heatmap. Black line marks the loess regression fit. Black points label the 52 differentially expressed genes (with a cutoff of FDR < 0.15) from **a** to **c**. **f** UMAP-based embedding of PV-INs from the CA1-IN study[10], and mapping the PV-INs of this current study onto the UMAP embedding using the extended set of $n = 52$ differentially expressed genes (with a cutoff of FDR < 0.15) from panel **a** to **c**. Three clusters of mapped cells are shown (labeled as 1, 2, and 3) as determined by K-means clustering. Symbols refer to the following transcriptomic subtypes, as described in the original study: rightward triangle *Pvalb.Tac1.Akr1c18*, leftward triangle *Pvalb.Clql1.Cpne5*, upward triangle *Pvalb.C1ql1.Npy*, downward triangle *Pvalb.Clql1.Pvalb*, plus sign *Pvalb.Tac1.Nr4a2*, open circle *Pvalb.Tac1.Sst*, and star *Pvalb.Tac1.Syt2*. The dashed line separates *Pvalb.Tac1* and *Pvalb.C1ql* types.

Next, we evaluated the expression of the 52 morphology-associated genes in the CA1-IN dataset. We found that the majority of genes, including these 52, were enriched in our data (Fig. 4e) and several of them were not or very rarely detectable in the CA1-IN dataset (2 genes were not detected, another 10 were detected in at most 20 out of 479 cells, and yet another 15 were detected in less than 10% of CA1-IN cells; Supplementary Fig. S7). This discrepancy, which may be due to the more than a 100-fold difference in alignment depths (10 million versus 0.1 million reads per cell in this and in the CA1-IN study), suggested a sub-optimal recovery of morphologically relevant genes in the CA1-IN study. Even with this caveat, UMAP dimension reduction using the 52 genes still introduced finer distinctions

within the CA1-IN *Pvalb.Tac1* and *Pvalb.C1ql1* types while preserving their global composition (Fig. 4f). Subsequent mapping of our PV cells onto this refined CA1-IN transcriptomic map leads to the following observations: (1) a part of *Pvalb.Tac1* was mapped by mostly BIC (vertical and horizontal) type cells (K-means "cluster 1" in Fig. 4f), (2) another part of *Pvalb.Tac1* was mapped by both BC (vertical and horizontal) and AAC, but not BIC, type cells ("cluster 2"), and (3) *Pvalb.C1ql1* were mapped by vertical AAC and BC, but not BIC, type cells ("cluster 3"). To summarize, our results revealed sub-optimal conditions to resolve all morphological PV types in this transcriptomic map. Nevertheless, they indicate that *Pvalb.C1ql1* represented cells with vertical dendrites, *Pvalb.Tac1* represented a mixed pool of vertical

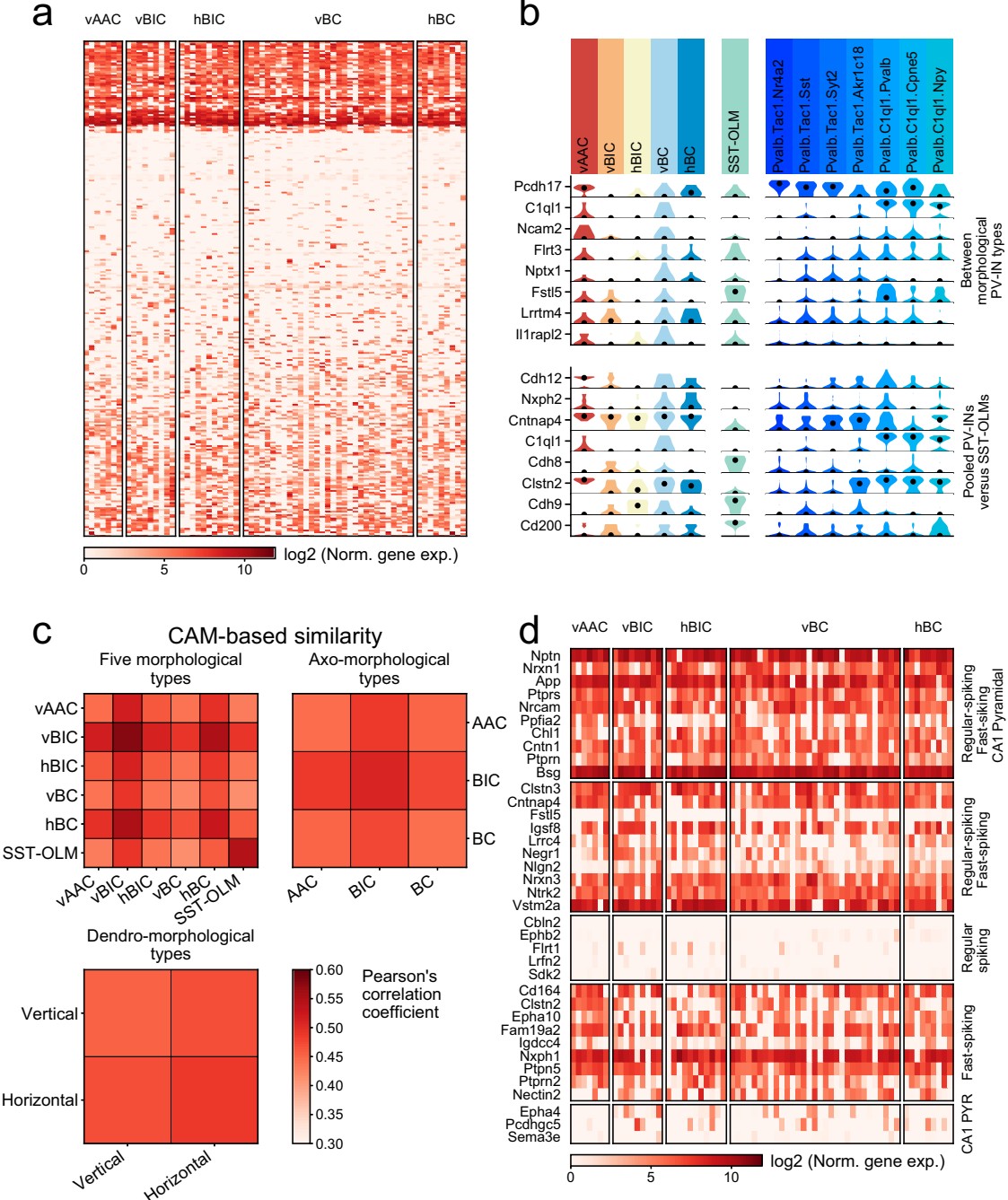

**Fig. 5 Analysis of CAM expression in morphological PV types. a** Heatmap showing all CAMs expressed in at least three PV-INs, grouped by morphological types. Genes are ordered via hierarchical clustering. **b** Violin plots show the top ten most significantly differentially expressed CAMs (using edgeR, fold difference >2), between morphological PV types (top panel), and between pooled PV-INs and SST-OLM cells (bottom panel). For comparison, relevant types from the CA1-IN study[10] are shown on the right. **c** Similarity matrices between the five morphological (top left), three axo-morphological (top right), and two dendro-morphological PV types (bottom left). Similarity scores were measured based on only CAM expression and using the average of Pearson's correlation coefficients of cells. **d** Heatmap shows an expression of genes that were previously described[13] to be (1) ubiquitous in CA1 pyramidal cells (PYR), fast-spiking interneurons (FS; presumed PV-INs), and regular-spiking interneurons (RS; presumed CCK-INs); or specifically expressed in (2) FS and RS cells; (3) RS cells; (4) FS cells; or in (5) PYR cells.

and horizontal type cells, and that the *Pvalb.Tac1.Sst* and *Pvalb. Tac1.Akr1c18* subregions ("cluster 1") appeared to consist of dominantly BIC-type cells.

**CAM expression among morphological PV types**. Although our results already revealed that transcriptomics only weakly predicts morphological PV-IN identities, we further investigated the expression of CAMs due to their importance in specifying neuronal connectivity and their close relation to neuronal identity. However, as predicted by the above analyses, CAM expression (based on 405 genes[13]) was uniform among the five morphological PV types (Fig. 5a). Although pairwise comparisons (using edgeR, fold difference >2, FDR < 0.05)

revealed fine distinctions between morphological PV types (Fig. 5b, upper panel), and between pooled PV cells and SST cells (Fig. 5b, lower panel), CAM-based similarity comparisons showed uniformity among the different PV types (Fig. 5c). Meanwhile, PV-INs consistently lacked expression of CAMs that were previously associated[13] with CA1 regular-spiking interneuron (RS; presumed CCK population) and CA1 pyramidal types (PYR; Fig. 5d), corroborating specific CAM expression when compared to developmentally different neuronal families. To sum, we conclude that CAM expression is uniform across the entire PV-IN population.

**Functional maturation of PV-INs.** To extend our analysis into an earlier developmental domain, such as the critical period of interneuron plasticity[36,38,39,41,42,47,54], we collected additional cells from younger than 21-day-old (<P21) animals. To make this analysis more focused, we emphasized collecting data from the vBC type, by patching tdTomato+ cells within the pyramidal cell layer of CA1, which consisted the majority of our >P21 dataset (in PV-Cre::Ai14 mice, PV and thus tdTomato+ expression first appears at ~P10, which limits cell collection from earlier time points using this approach). In this manner, we complemented our existing dataset with an additional $n = 19$ vBCs and analyzed a combined number of 50 vBC type PV-INs collected between P10 and P77.

Morphological analysis showed that most P10–20 vBC type PV-INs already display fully developed axonal and morphologic features (Fig. 6a). Sholl analysis on dendritic arborizations showed similar patterns between <P21 and >P21 cells (Fig. 6b), and dendritic and axonal lengths were not significantly different ($P = 0.06$ and 0.78, respectively; Fig. 6c, d), where P21 was used as an arbitrary cutoff. However, electrophysiological properties displayed age-dependent changes. Specifically, we found an increase in AP amplitude (FDR = 0.049) and a decrease in AP half-width (FDR = 0.0082), in >P21 versus <P21 cells. Furthermore, we found a smaller AP firing attenuation (FDR = $1.7 \times 10^{-5}$, two-sided Mann–Whitney test was used for these comparisons; Fig. 6e) in >P21 versus <P21 cells. While these correlated with age (Supplementary Fig. S8), physiological parameters could not separately cluster differently aged PV-INs (Fig. 6f). To elaborate in the transcriptomic domain, we examined ion channel-coding genes, and identified only one significant change, the increased expression of the GABA receptor subunit *Gabra1* in P > 21 (Supplementary Fig. S9).

**Rapid transcriptomic changes in PV-INs between P21 and P25.** Using two independent bioinformatic approaches, we then examined whether transcriptomic changes other than ion channels corresponded to PV cell maturation. First, we used a sliding-window approach, only considering whether a gene was expressed or not, independent of its expression level (Fig. 7a and "Methods"). Second, we used Monocle[55], which relies on expression levels and calculates the cells' pseudo time best correlating with age, and finds genes whose expression level significantly correlates with this (Supplementary Fig. S10). As a result, we found a short period between P21 and P25, which was marked by pronounced downregulation of $n = 6$ and upregulation of $n = 4$ genes (Monte Carlo on Gini impurity, $P < 0.1$; "Methods"; Fig. 7a). This pattern was robust and separately clustered cells by their age (Fig. 7b) and moreover, using random forest classification, predicted whether the animals' age was below P21 or above P25 (Fig. 7c). During this transition period, *Rfxank*, *Dpysl5*, *Timmdc1*, *Zfp629*, *Tbc1d9*, and *Fam181b* were rapidly downregulated (Fig. 7a).

**The onset of hemoglobin mRNA expression in PV-INs.** Among the upregulated genes we found *Gh* (or growth hormone 1), and *Hba-a1*, *Hbb-bt*, and *Hbb-bs*, which are all hemoglobin (Hb) subunit-coding genes. Although not detected by our initial analysis, we also examined the expression of *Hbb-a2* (another key component of functional Hb tetramers) and that of *Mg* (myoglobin), *Ngb* (neuroglobin), and *Cytb* (cytoglobin), which all display oxygen-binding properties similar to Hb[56]. *Hbb-a2* followed a similar expression pattern as the other Hb-coding genes. By contrast, *Mg* and *Ngb* were not expressed, whereas *Cytb* was highly expressed, without regard to age (Fig. 7d, e and Supplementary Fig. S10). We also explored the expression of genes that are known to regulate Hb expression. We found a lack of canonical Hb transcriptional regulators GATA-1 and -2[56,57], but stable (age-independent) expression of *Hif1a*, a hypoxia response-related transcriptional factor that also controls Hb expression (Supplementary Fig. S10). This onset of Hb expression was characteristic to all hippocampal PV, but not SST, types (Fig. 7f and Supplementary Fig. S11). Finally, while the analysis of additionally collected cortical >P21 tdTomato+ cells ($n = 6$; Fig. 7f) and bioinformatic analysis of other datasets (Supplementary Fig. S11) revealed a lack of regular hemoglobin-related gene expression in cortical PV cells, nucleotide level analysis of hippocampal PV types (Fig. 7g and Supplementary Fig. S12) and ISH staining of Hb subunits from the Allen Mouse Brain Atlas (Supplementary Fig. S10) provided further support for Hb mRNA expression in hippocampal PV cells.

## Discussion

Multiple studies already suggest that physiological features can be inferred from single-cell transcriptomic data[12–17,19–21,45]. By contrast, on-going efforts strive to infer neuronal morphology from transcriptomic information (in somatosensory[14], visual[20,21,58,59], and motor cortex[60]). Large-scale transcriptomic assays have previously described continuously varied gene expression among and within cell types, in which clusters may split (discreteness) or merge (continuous variation), depending on gene detection, cell sampling numbers, and noise estimates, or statistical criteria[11]. For CA1 interneurons specifically, continuous variation was earlier suggested based on a transcriptomic map that also included two distinct transcriptomic PV-IN clusters, appointed to be presumed AAC and BC/BIC types[10]. Here, we sequenced mRNA from morphologically identified PV-INs in hippocampal CA1 to relate transcriptomic content to morphology and circuit connectivity.

**Transcriptomic correlates of morphological PV types.** Using unsupervised feature selection and dimension reduction on the cells' whole transcriptome, we showed that while PV and SST cells can be transcriptionally separated into their respective populations, known morphological PV types could not be accurately distinguished. Four transcriptomic cell types could be defined using proMMT[10], in which *Pthlh* and *Synpr* were the strongest separators. Of these proMMT types, BIC types largely overlapped with the *Synpr* proMMT type whilst AAC and BC types intermingled within the remaining three *Pthlh* proMMT cell types. Meta-analysis of the CA1-IN dataset with our own revealed that our cells corresponded to the presumed PV population, but did not support the hypothesis that morphological AAC and BC/BIC types defined major transcriptomic types (Figs. 1–3). Supervised gene selection based on morphological types reproduced some genes that were differentially expressed between transcriptomic types (*Npy*, *Sst*, *Synpr*, *Pthlh*, *Zcch12*, *Pde11a*, *C1ql1*, *Cd36*, and *Mctp2*) and yielded known and novel subtype-specific gene expression patterns (Figs. 2 and 4).

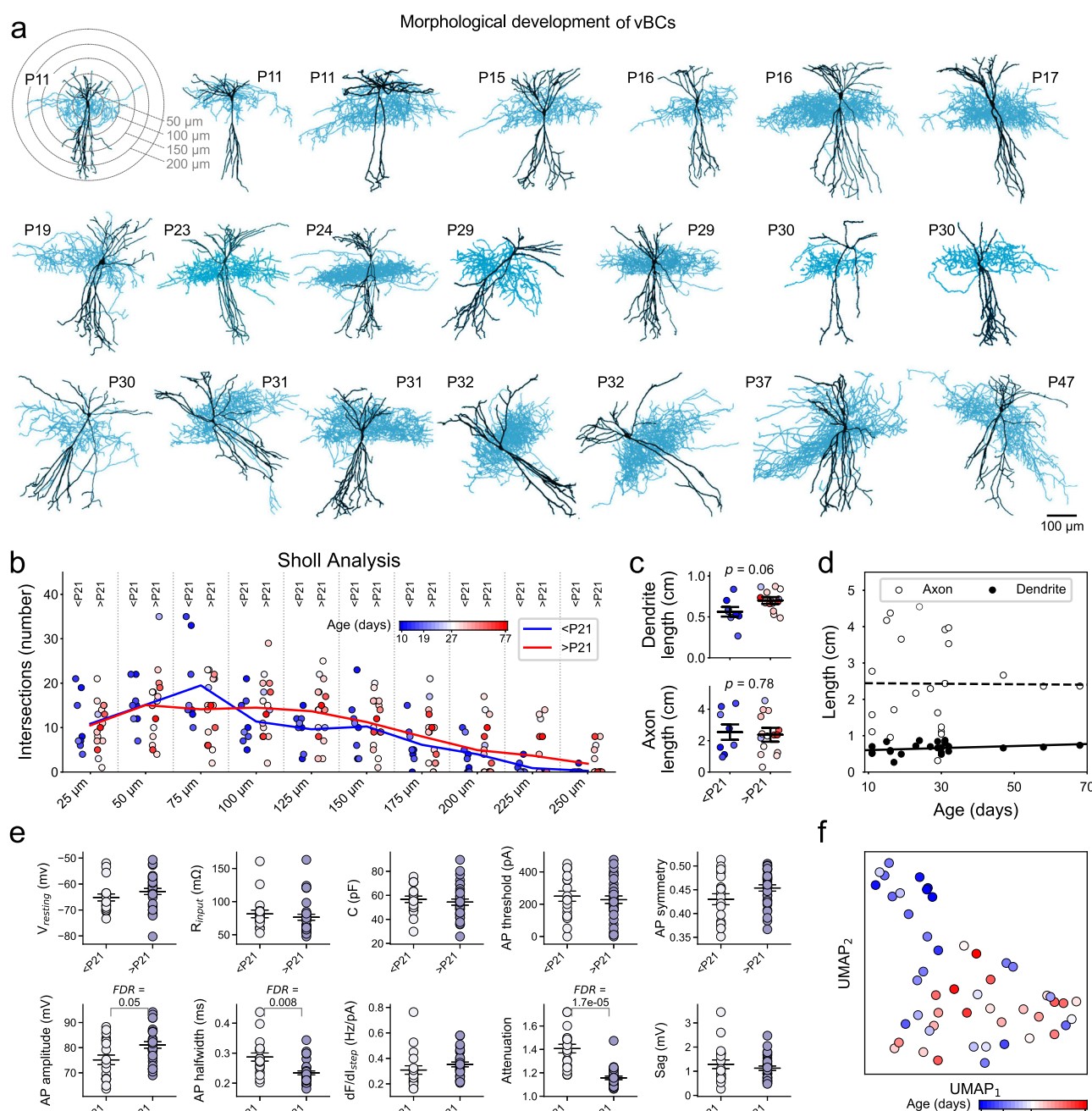

**Fig. 6 Morphological and electrophysiological analysis of vBC type PV-INs during circuit maturation. a** Morphological reconstructions show vBC type PV-INs at different ages. The first image shows Sholl analysis for dendritic branching based on concentric 3D spheres (plot shows circles for clarity). **b** Plot shows the number of intersections at different distances from the soma of $n = 8 < P21$ and $n = 16 > P21$ vBC. Circles denote individual cells, which represent a randomly chosen subset (~50%) of all <P21 and >P21 vBC cells included in the whole study. None of the statistical comparisons (two-sided Welch's $t$ test) revealed a FDR smaller than 0.255. **c** Plots show the total dendritic (left) and axonal (right) length of <P21 and >P21 vBC-type cells. $P$ values are shown on top and were determined using two-sided Welch's $t$ test. Data represent mean ± S.E.M. **d** Scatter plots and linear regression fits of axon and dendrite length against age. Neither value was shown to correlate with age (lowest $P$ value and highest $r^2$ for fit were 0.272 and 0.209, respectively). **e** Plots show electrophysiological properties of $n = 19 < P21$ versus $n = 31 > P21$ vBC type PV-INs. Attenuation (FDR $= 1.7 \times 10^{-5}$, two-sided Welch's $t$ test), AP half-width (FDR $= 0.008$) and AP amplitude (FDR $= 0.05$) show significant changes. Data represent mean ± S.E.M. **f** UMAP-based dimension reduction of electrophysiological properties of vBC-type cells. All cells represent the vBC type and are colored by age.

As known markers, *Sst* and *Npy* were previously reported to be enriched in the BIC type[61]. As a finer distinction, our data also revealed that *Npy* was enriched in vBIC, whereas *Sst* was more enriched in hBIC. As novel markers, *Synpr, Akr1c18, Zcchc12*, and *Stxbp6* were enriched in the BIC type, whereas among others

*Pthlh, Pcdh17*, and *Kcng4* and were enriched in the AAC and BC types. Of these, (1) the correlation of *Akr1c18* with fast- and delay-spiking markers within the *Pvalb.Tac1* population has been highlighted, but without association to morphology[10]. (2) *Kcng4*, which encodes a modulatory subunit for the potassium channel

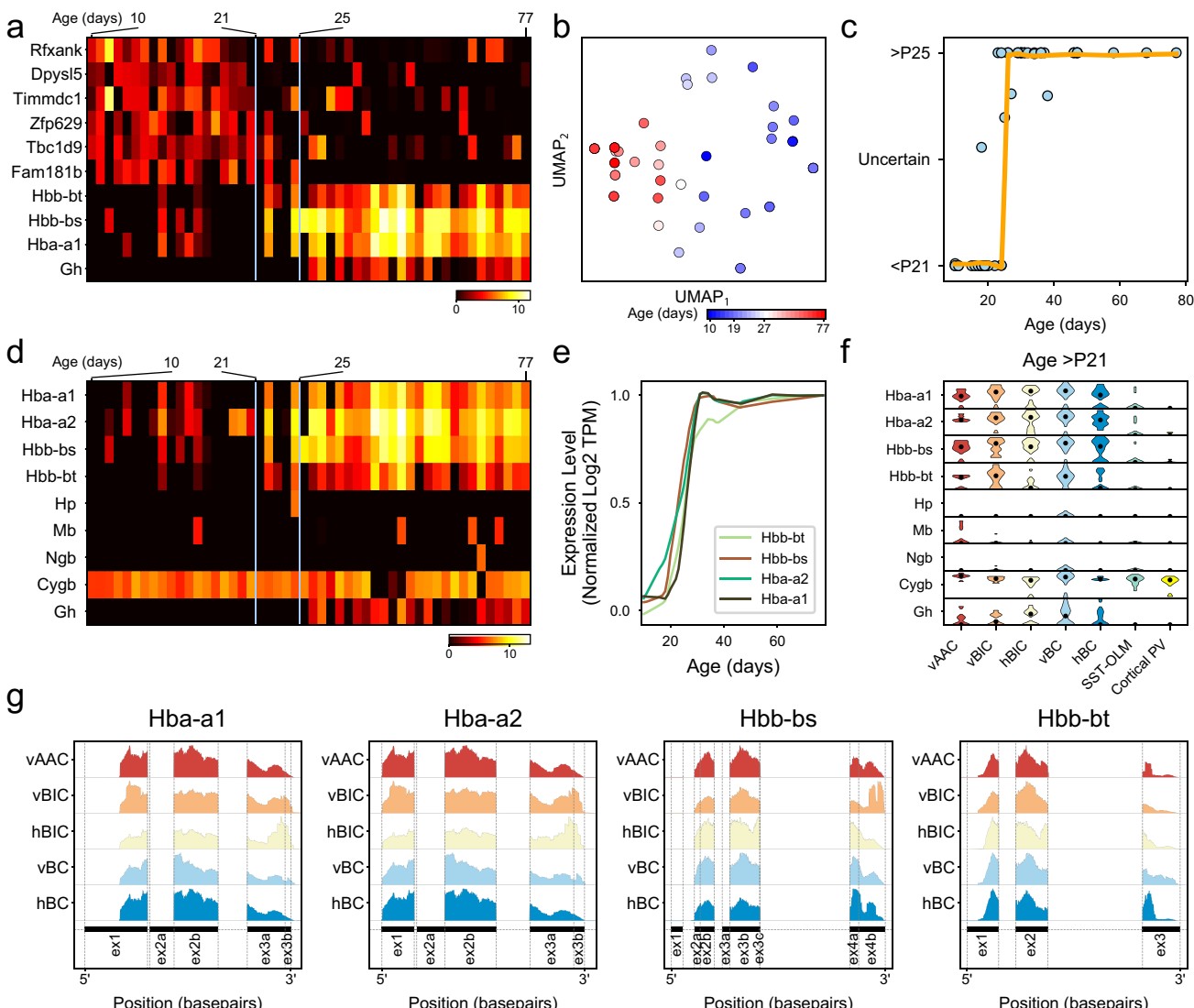

**Fig. 7 Age-dependent transcriptomic changes and the onset of hemoglobin mRNA expression in PV-INs. a** Expression levels of genes showing a statistically significant (FDR < 0.10) transition using Gini impurity (see "Methods"). Genes (rows) ordered by the day of transition, cells (columns) are ordered according to age. All cells represent vertical basket (vBC) type. **b** UMAP-based on genes from panel **a**. All cells represent vBC type and are colored by age. **c** Plot of consistency with which age can be predicted using Random Forest Classifier based on genes from panel a (see "Methods"), fitted with a loess curve. **d** Heatmap shows the expression of hemoglobin mRNA and related genes in vBC-type cells (columns), which are sorted by age left to right. **e** Normalized loess fits of hemoglobin mRNA expression levels versus age in vBC-type cells. **f** Violin plots show expression of hemoglobin mRNA and related genes from panel **d** only considering >P21 PV-INs and SST-OLM cells. **g** Plots show average *Hba-a1*, *Hba-a2*, *Hbb-bs*, and *Hbb-bt* expression in each morphological PV type with single-nucleotide resolution. Exon and intron lengths are shown according to their original scale.

Kv2.1[62], may represent a highly selective marker for both AAC and BC types. According to the Allen Mouse Brain Atlas, *Kcng4* is present only in a handful of cells in the pyramidal layer of CA1, plausibly suggesting restricted expression in PV-INs. Added to this, our data shows specificity to AAC and BC within PV-INs. (3) *Pthlh* expression was previously found to correlate with fast-spiking property of PV-INs in the dorsal striatum[15]. By contrast, our data in CA1 show a correlation of *Pthlh* with morphological features. Since the striatum study did not include morphological characterization, it is possible that fast-spiking property in striatal PV-INs also correlated with morphology. Although our data revealed multiple genes that differentiated BIC from AAC and BC types for both supervised and unsupervised analysis, it did not disclose genes differentiating the AAC and BC types. This contrasts with a previous observation made in the CA3 area,

where SATB1 was specifically expressed in the AAC, but not BC, type[63].

Our results shed light on transcriptomic differences among dendro-morphological types. We identified 14 and 2 genes, which were selectively enriched in vertical and horizontal types, respectively, without regard to the cells' axo-morphological features. This suggests that dendro-morphological features may need to be considered when interpreting transcriptomic types. Meta-analysis of the CA1-IN dataset[10] with our own revealed that *Pvalb.C1ql1* likely represented cells with vertical dendrites (we detected *C1ql1* only in vertical, but not in horizontal, cells, Fig. 4), whereas *Pvalb.Tac1* represented a mixed pool of cells with vertical and horizontal dendritic cells. In conclusion, our analyses showed that while the above-highlighted marker genes can differentiate the BIC and AAC/BC types from one other, the transcriptome of

morphologically different PV types display a high degree of similarity with >99% of the genes are not differentially expressed in any pairwise comparison. As a consequence, single-cell mRNA profiles remain a weak predictor of PV-IN morphology. In agreement with this, other multimodal cell-classification studies (in visual[58,59] and motor[60] cortex) reached similar conclusions. The one exception to the agreement between these studies is that cortical chandelier (or AAC) cells formed a separate cluster in the transcriptomic space ("*Pvalb_Vipr2_2*" cells in Scala et al.[60]), whereas in our study hippocampal AACs did not.

**Molecular architecture of circuit connectivity**. Transcriptomic data collected from morphological PV types allowed us to further test a key theory of brain connectivity which states that synaptic CAMs specify neural connections. While ample evidence supports this hypothesis[28–30,32], in seeming contradiction, our data revealed similarity in CAM expression among the morphological PV types sampled from P21 and older mice (Fig. 5). However, this outcome was not completely unexpected. Transcriptomic studies have revealed major CAM differences between different neuron families, such as between excitatory versus inhibitory cells or between inhibitory cells with different developmental origin[11,13,35], but CAM diversity appeared to be less pronounced when only cells within individual neuron families, such as PV-INs, were considered[13,35]. However, supporting evidence demonstrating the cells' distinct morphology or connectivity was lacking from these studies. Our current study provides evidence for virtually identical CAM profiles among PV-INs, in spite of differences in their morphological features. As a corollary of this finding, just as transcriptomic information is a poor predictor of cellular morphology, our ability to use gene-level mRNA profiling to infer circuit connectivity remains limited. Nevertheless, our data do not exclude the hypothesis that input/output connectivity of these cells is defined by CAMs. First, it is possible that isoform level, non-mRNA, or translational information is instead more predictive of morphology and circuit connectivity[22]. Second, CAMs could specify connections exclusively during earlier development, after which key specification factors are downregulated[64], and CAM function shifts toward the maintenance and modulation of synaptic transmission. Finally, since the major postsynaptic target of all PV types are the CA1 pyramidal cells, axo-morphological distinctions among the different PV types define sub-cellular zones where PV synapses are established on pyramidal cells. Therefore, it remains plausible that the observed uniform CAM profile is responsible for target cell-type selectivity, i.e., CA1 pyramidal cells, but does not account for sub-cellular targeting differences.

**Switch of transcriptomic states and rapid onset of Hb expression**. During the second postnatal week of cortical development, fast-spiking interneurons display intense morphological, physiological[38,39,41], and transcriptomic changes that involve thousands of genes[45]. Ion channel-coding genes change their expression with up to 10–100-fold magnitude, which coincides with the profound electrophysiological maturation of cells. By contrast, our analysis did not register electrophysiological, transcriptomic, or morphological changes at a similar scale, suggesting that, in hippocampal CA1, intrinsic maturation of PV-INs is largely completed by P10. More specifically, a previous report on the development of fast-spiking PV cells in the dentate gyrus revealed major morphological and biophysical changes in cells ranging from P4 to P22[38]. Starting our dataset at a later time point, we found that P10–P21 vBC-type PV-INs display developed axonal and dendritic features. We also detected an increase in AP amplitude and smaller AP firing attenuation. Furthermore,

we found a decrease in AP half-width and there was, though not significant, an increasing trend in the dF/dI, which are in line with decreasing AP half-width and increasing maximum firing frequency reported in developing dentate gyrus PV cells[38]. Regardless of any biophysical disparities, our transcriptomic data did not display changes in voltage-gated ion channel-coding genes. However, it revealed another wave of transcriptomic regulation, which occurred at a later time window (between P21 and P25) and was restricted to a smaller number (ten) of genes, either down- or upregulated (Fig. 7). Importantly, these genes did not include CAMs, suggesting that any potential downregulation within this gene family occurred during earlier development, in accordance with a seeming morphological completion after P10 (Fig. 6).

During the transition time window, occurring between P21 and P25, six genes were rapidly downregulated. Of these, *Rfxank*, *Dpsys15*, *Tbc1d9*, and *Fam181b* have been previously linked to different stages of neural development[65–68]. Upregulated genes encoded Hb subunits. While this was unexpected, Hb expression has been previously demonstrated in a limited number of neuronal types, including A9 dopaminergic[69] and unidentified types of cortical, hippocampal, and cerebellar cells[70–72]. Furthermore, ISH staining in the Allen Mouse Brain Atlas supports the neuronal expression of multiple Hb subunits in the hippocampus, cortex, and throughout the adult brain (Supplementary Fig. S10 and Allen Mouse Brain Atlas).

In our data, the onset of Hb mRNA expression characterized all CA1 morphological PV types. The role of Hb in neurons remains controversial. In hypoxia, Hb appeared to be neuroprotectant by rendering cells into an oxygen privileged state[71]. However, neurodegenerative effects of Hb expression were also proposed (in aging[73], by promoting Aβ oligomerization[70], by toxic Hb aggregate formation[72], and in learning impairments[74]). In the hippocampus specifically, chronic stress led to a significant downregulation of Hb genes[75], whereas early-life iron deficiency anemia altered the development and long-term expression of parvalbumin and perineuronal nets[54]. While our results do not clarify the role of Hb gene expression in neurons, nor if these transcripts are subject to translation, they make an observation, which suggests that PV-INs may be a cellular substrate behind Hb-associated network effects.

**Summary**. This study performed a multimodal analysis of hippocampal PV-INs in the electrophysiological, morphological, and transcriptomic domains. Outcomes identified genes that differ between morphological PV types but overall corroborated a high degree of transcriptomic similarity across the entire PV population. Furthermore, this study provides evidence for a lack of differentiating CAMs (as defined by mRNA-based, gene-level transcriptomic readout) in cell types that have different wiring patterns. Finally, the results of this study demonstrate a switch of transcriptomic states and rapid onset of Hb mRNA expression whose functional relevance remains unclear.

## Methods

**Animals**. All animal protocols and husbandry practices were approved by the Veterinary Office of Zürich Kanton. The University of Zurich animal facilities comply with all appropriate standards (cages, space per animal, food, water), and all cages were enriched with materials that allow the animals to exert their natural behavior. Mice were housed at room temperature, 12–12 h light cycle, and at 40–60% humidity. Both males and females were used for all experiments. Animals were sacrificed from P10 and older. The offsprings (Pv-Cre::Ai14) from the breeding between the following two lines were used in this study: (1) Pv-Cre: Pvalb(cre) > <GM > < B6;129P2-Pvalb<tm1(cre)Arbr > /J(#008069) and (2) Ai14: Ai14 > <GM > < B6.Cg-Gt(ROSA)26Sor < tm14(CAG-tdTomato)Hze > /J (#007914).

**Electrophysiology**. Hippocampal slices (300-μm thick) were prepared from P10 and older mice, and incubated at 34 °C in sucrose-containing artificial cerebrospinal fluid (sucrose-ACSF) (85 mM NaCl, 75 mM sucrose, 2.5 mM KCl, 25 mM glucose, 1.25 mM NaH$_2$PO$_4$, 4 mM MgCl$_2$, 0.5 mM CaCl$_2$, and 24 mM NaHCO$_3$) for 0.5 h, and then held at room temperature until recording. Cells were visualized by infrared differential interference contrast optics in an upright microscope (Olympus; BX-51WI) using a Hamamatsu Orca-Flash 4.0 CMOS camera. Recordings were performed using borosilicate glass pipettes with filament (Harvard Apparatus; GC150F-10; o.d., 1.5 mm; i.d., 0.86 mm; 10-cm length) at 33 °C in ACSF (126 mM NaCl, 2.5 mM KCl, 10 mM glucose, 1.25 mM NaH$_2$PO$_4$, 2 mM MgCl$_2$, 2 mM CaCl$_2$, and 26 mM NaHCO$_3$) with a standard intracellular solution (95 mM K-gluconate, 50 mM KCl, 10 mM HEPES, 4 mM Mg-ATP, 0.5 Na-GTP, 10 mM phosphocreatine; pH 7.2, KOH adjusted, 300 mOsm). All recordings were made using MultiClamp700B amplifier (Molecular Devices), and signals are filtered at 10 kHz (Bessel filter) and digitized (50 kHz) with a Digidata1440A and pClamp10 (Molecular Devices).

**Identification of cell types**. Neurons were identified by fluorescent labeling in hippocampal brain slices prepared from Pv-Cre::Ai14 mice. Fluorescence-labeled cells were variably present in all hippocampal strata. During recording, cells were filled with biocytin (Sigma-Aldrich, 2%) for subsequent post hoc visualization of axons. After collection of cytosols, brain slices were fixed in 4% paraformaldehyde (Sigma-Aldrich) overnight and subsequently processed for immunostaining with streptavidin-alexa Fluor 488 conjugate (Invitrogen, Thermo Fisher Scientific). Only those cells were included, where staining revealed axonal and dendritic arborization. Out of 340 ($N = 110$ are <P21, $N = 224$ are >P21) hippocampal PV cells recorded for the whole study, 123 cells were further considered for sequencing based on sufficient staining of the axons and dendrites. For morphological reconstructions, we used a confocal microscope to create image stacks and applied the "simple neurite tracer" function in Fiji ImageJ to trace axons and dendrites of cells. In some cases, the cutting angle used for brain slicing caused the CA1 pyramidal layer, and thus all other layers, to be non-perpendicular to the slice. In such case, viewed from the angle of imaging, different layers overlap with each other. This was particularly relevant for cells that could be perceived as trilaminar cells, yet were in fact BIC-type cells when viewed from an angle perpendicular to the CA1 cell layers. For such angular adjustment, we applied the "3D project" function in Fiji ImageJ. Of these, 88 cells, including 67 that were >P21 cells and 21 that were <P21 cells, were morphologically classified and passed quality control, 14 cells could not be unambiguously classified as either of the five PV types and 18 cells did not pass quality control after single-cell RNA sequencing. In addition, six further PV cells were collected from the cortex, which was sequenced but not morphologically analyzed. Cells are listed by cell name, cell type, age in the Supplementary Information.

**mRNA sample collection**. To minimize interference with subsequent molecular experiments, only a small amount of intracellular solution (~1 μl; not autoclaved or treated with RNase inhibitor) was used in the glass pipette during electrophysiological recordings[13,18]. Before and during recordings, all surface areas—including manipulators, microscope knobs, computer keyboard, etc.—that the experimenter needed to contact during experiments were cleaned with RNase Away solution (Molecular BioProducts). After recordings, the cell's cytosol was aspirated via the glass pipette used for recording. Although the aspirated cytosol may have contained genomic DNA, our choice of cDNA preparation, which involved poly-A based mRNA selection, eliminates the possibility of genomic contamination in the RNA-seq data. For sample collection, we quickly removed the pipette holder from the amplifier head stage and used positive pressure to expel samples into microtubes containing cell collection buffer while gently breaking the glass pipette tip. Cell collection microtubes were stored on ice until they were used.

**cDNA library preparation**. Single-cell mRNA was processed using Clontech's SMARTer Ultra Low RNA Input v4 or SMART-Seq HT kit. As the first step, cells were collected via pipette aspiration into 1.1 μL of 10× collection buffer and spun briefly before they were snap-frozen on dry ice. Samples are stored at −80 °C until further processing, which was performed according to the manufacturer's protocol. The resulting cDNA was analyzed on the Fragment Analyzer (Advanced Analytical). Library preparation was performed using the Nextera XT DNA Sample Preparation Kit (Illumina) according to the manufacturer's protocol. Following library preparation, cells were pooled and sequenced using NextSeq 300 high-output kit in an Illumina NextSeq 500 System with 2 × 75 paired-end reads.

**Processing of RNA-seq data**. After sequencing, raw sequencing reads were aligned to the Ensembl GRCm38 reference transcriptome (Version 95), using Kallisto's *quant* command[76] with 100 bootstraps. For convenience, Ensembl gene IDs were converted to gene symbols using a reference file generated by biomaRt[77]. In the few cases where different Ensembl gene IDs identified the same gene symbol, transcript per million (TPM) levels were summed.

**Sequencing data quality control**. All data analysis was performed using R and Python codes. First, in each cell, we calculated the number of unique genes and the number of aligned reads. Second, we calculated the median and median absolute deviation of these two values across all cells. Cells that had either value more than three median absolute deviations below the median were removed as failing quality control.

**Differential gene expression analysis**. Transcript per million (TPM) normalization of transcripts was calculated by a built-in Kallisto function. For calculating differentially expressed (DE) genes, we first read in Kallisto's output using Tximport[78], to account for uncertainty in alignment. We then imported the results to edgeR and used a quasi-likelihood test on all genes that were expressed at a level of TPM > 15 in at least five cells in the two groups being compared (resulting in between 7033 and 8568 genes considered out of 35,825 total). Genes were labeled as DE if there was a fold difference of at least 2 (absolute value of logFD>1) in average expression, at a significance of $P$-adjusted <0.05.

**Dimension reduction methods and analysis**. To plot high-dimensional data, we used five dimension reduction algorithms: principal component analysis (PCA), t-distributed stochastic neighbor embedding (t-SNE), Fast Fourier Transform-accelerated Interpolation-based t-SNE (FIt-SNE), negative binomial t-SNE (nbt-SNE), and uniform manifold approximation and projection (UMAP). All methods transform high-dimensional data to a lower dimension while preserving key information. PCA is a linear transformation that attempts to preserve the variance in the positions of cells. t-SNE is a nonlinear transformation that attempts to preserve the distances of cells only to their nearest neighbors, losing macroscale information in the process. Both FIt-SNE and nbt-SNE are modifications to t-SNE. FIt-SNE is designed to make t-SNE run faster on large-scale data and attempt to preserve macroscale information, while nbt-SNE modifies the distance function used from a Gaussian to a Negative Binomial model that is believed to be more accurate for RNA-sequencing data. UMAP works similarly to t-SNE, but uses a modified distance function and attempts to preserve macroscale information by putting more weight on distances between farther away points. While PCA is unable to capture more complex, nonlinear information, it has the advantage of interpretability; only on a PCA plot do the distances along each axis have any biological meaning.

**Classification accuracy**. Related to Figs. 1, 6. To determine how accurately cell types could be classified, we trained a Random Forest Classifier algorithm. Briefly, a Random Forest creates multiple decision trees to classify cells, using an only a different subsets of the genes each time. Each of the individual trees "votes" on classification and the most popular classification is used. We used an ensemble of 100 decision trees for our algorithm. To not bias the results of the Random Forest, and to simplify interpretation, we randomly removed cells from the larger class so that the two categories had the same number of cells. This allowed us to label 50% accuracy as the baseline of what we would get if there were absolutely no differences between the two categories. We then used 80% of the cells as a training set and evaluated the result on the remaining 20%. For each classification we repeated this method 100 times, each time randomly selecting which cells were removed, and which were used in the training and test set. This allowed us to get an average classification accuracy for the categories as a whole, as well as for each cell.

**Gene selection**. Related to Fig. 2: In a bioinformatics dataset, when exploring the difference between multiple cell types, or trying to identify cell types via clustering, most genes do not contribute any information to the separation. Continued consideration of these genes can decrease the signal-to-noise ratio to the point where existing distinctions cannot be resolved. As such, it is important to first trim the list of genes to only keep significant genes. However, there is no singular "best" approach to select these genes for specific types of problems, let alone all bioinformatics analysis in general. As such, we tried a number of different gene selection methods to confirm if any of them would give a clear separation. Three of them (chi-squared, mutual information, and ANOVA F-value) were already implemented in scikit-learn. For chi-squared, we used log2 of gene TPM expression levels, while for mutual information and ANOVA F-value we used a boolean value of whether or not a gene was expressed. In all three cases, we took the best separating 150 genes. Next, we tried the 150 genes that were used for classifying the CA1-IN data. We also tried the top 150 genes that correlated with these separators but were not them. Lastly, we ran a method described by Kobak et al.[79] that finds the most relevant variable genes accounting for expression rate, and using a cutoff of TPM > 32, and once again took the top 150 genes.

**Transcriptomic mapping**. Related to Figs. 2, 4: To map our data onto the CA1-IN dataset, we used the method described in Kobak et al.[79] Briefly, after key genes were selected, we used the correlation coefficient to find the k-nearest neighbors for each cell. We took the median position of the embeddings of these k neighbors, as the mapping position of our cells.

**Sliding window**. Related to Fig. 7: To determine whether the expression of the gene was "upregulated" or "downregulated" at a given age, we only considered gene expression data in a binary format, i.e., expressed or not in a single cell. Since

Kallisto's bootleg approach has shown to occasionally assign very low expression levels to transcripts that are not expressed, we used a low cutoff (0.6 TPM) for determining if a gene was expressed or not. To increase the statistical power, we ignored genes that were expressed in less than six cells, or not expressed in less than six cells. After that, for each gene, we used a sliding window to calculate the transition point with the highest loss of Gini impurity. To calculate the P values, we used a Monte Carlo simulation. For each potential number of cells that a gene might be expressed in, we ran 100,000 simulations by randomizing the expressions and calculated the Gini Impurity loss for each. We then used these distributions to calculate the P value for each gene.

**Gini impurity**. Related to Fig. 7: Gini impurity is a measure of the degree of heterogeneity of a group. If elements in a set were randomly labeled based on the distribution of categories in a set, the fraction that would be incorrectly labeled is the Gini impurity. It can be calculated as $1 - \text{Sum}(p_i^2)$, where $p_i$ is the fractions of a set that belong to each group. The number varies from 0 (complete uniformity) to almost 1 (every element is in a different group), and from 0 to 0.5 when there are only two groups. If a set is divided into two smaller parts, the average Gini impurity—normalized by the sizes of the two subsets—of the two subsets will be smaller than that of the entire set. This difference is a measure of the information gain from the separation.

**Linear support vector machine with recursive feature elimination**. Related to Supplementary Fig. S6: Support vector machines are a classification algorithm. They find the best line (two features), plane (three features), or hyperplane (four features and above) along which to separate the data for classification. Linear support vector machines use features as is, rather than generating new features, making them simple to interpret. Recursive feature elimination is a way to reduce the number of features used in a classification algorithm. Briefly, after a classification algorithm is trained, the features are weighted, and the least important features are dropped. These two steps are repeated until the dataset is reduced to a previously selected number of features. For our data, genes were the features, and cell types were the classes that we tried to separate. Using the two algorithms together, we found the top 50 genes that best-classified cell types using a linear support vector machine.

**Electrophysiological analysis**. Related to Figs. 3, 6, and Supplementary Fig. S5: For electrophysiological analysis, to keep our data consistent, we only considered cells that passed transcriptomic quality control. We calculated and used a total of ten parameters. Voltage-Clamp Recordings: We applied a 5 mV voltage step to measure the resulting current amplitude over time. Input resistance was calculated from the resulting current amplitude at the steady-state. Series resistance was calculated from the same voltage step using the initial maximum current amplitude at the beginning of the voltage step. Capacitance was determined using the charge of the 5 mV voltage step (i.e., the area between the steady-state current response, input resistance, and the initial current response series resistance), and divided the charge by the voltage to reveal the capacitance of the cell. Current-Clamp Recordings: We applied current injections in steps (traces) overtime periods of 1.5 s, and measured the spiking response from the cells. Firing threshold was calculated by measuring the action potential (AP) spiking frequencies at each trace, and fitting the frequency vs current curve to a sigmoid function. We fit 20–80% (amplitude) of the sigmoid function with a linear function and judged the intersection of the fit with the "x" (current injection)-axis as the firing threshold. We determined change in firing frequency as a function of current injection ($dF/dI_{step}$) as the slope of a similar linear fit of the data, starting at 50 pA after the first trace with spiking, and ending 250 pA later. Attenuation was measured as the ratio of the first and last AP peak amplitudes at the maximal recorded trace. The amplitude of Sag potential was measured in response to −150 pA current injections and represents the difference in minimum membrane potential measured between 750 ms and 1000 ms and median membrane potential between 1500 ms and 2250 ms during the current injection. We analyzed single AP properties by averaging all time-aligned APs from an individual trace that contained at least three APs. These averaged APs were fit with three linear regressions in three separate time intervals: (i) baseline fit before AP, (ii) ascending (20–80%) phase of AP, and (iii) during the descending phase of AP (80–20%). In addition, we determined AP peak (max) and trough (min) times and voltage amplitudes. Using these readouts, we determined AP base width, AP half-width, as well as AP symmetry (i.e., temporal position of the peak between ascending and descending phases of AP).

**Reporting summary**. Further information on research design is available in the Nature Research Reporting Summary linked to this article.

## Data availability

Single-cell RNA sequencing data generated in this study have been deposited in NCBI GEO with the accession code GSE142546. Datasets analyzed during this study are available in the NCBI GEO repository: Winterer et al.[18]: https://www.ncbi.nlm.nih.gov/geo/query/acc.cgi?acc=GSE124847, Földy et al.[13]: https://www.ncbi.nlm.nih.gov/geo/query/acc.cgi?acc=GSE75386, Cadwell et al.[12]: https://www.ebi.ac.uk/arrayexpress/experiments/E-MTAB-4092/, Muñoz-Manchado et al.[15]: https://www.ncbi.nlm.nih.gov/

geo/query/acc.cgi?acc=GSE119248, Fuzik et al.[14]: https://www.ncbi.nlm.nih.gov/query/acc.cgi?acc=GSE70844, Zeisel et al.[8]: https://www.ncbi.nlm.nih.gov/geo/query/acc.cgi?acc=GSE60361, Harris et al.[10]: https://www.ncbi.nlm.nih.gov/geo/query/acc.cgi?acc=GSE99888, Tasic et al.[11]: https://www.ncbi.nlm.nih.gov/geo/query/acc.cgi?acc=GSE115746. All data supporting the findings of this study are provided within the paper and its Supplementary Information. All additional information will be made available upon reasonable request to the authors. Source data are provided with this paper.

## Code availability

The code used in this study is available from https://github.com/foldy-lab/Transcriptomics-of-Pvalb-Cells.

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

## Acknowledgements

We thank Drs. Gabe Murphy, János Szabadics, and Jochen Winterer for discussions. This work was supported by funding from the Swiss National Science Foundation (Switzerland, CRETP3_166815 and 31003A_170085) and the Dr. Eric Slack-Gyr-Stiftung (Switzerland).

## Author contributions

L.Q. performed electrophysiological and morphological experiments. L.Q. and W.L. performed single-cell RNA-seq. D.L. performed bioinformatic analyses. C.F. wrote the paper. C.F., D.L., and L.Q. designed the study, analyzed the data, prepared figures, and edited the paper.

## Competing interests

The authors declare no competing interests.
