## [Peer Review File · Nature Communications]

Reviewers' Comments:

Reviewer #1:

Remarks to the Author:

This study applied patch-seq to study the correlation between morphology, electrophysiology and gene expression in parvalbumin-positive (PV+) interneurons in CA1 of the hippocampus. The major findings are that 1) there does not seem to be a very good correlation between morphologic subtypes of PV+ neurons, transcriptomic cell types, and electrophysiological properties; 2) despite their different morphologies, different morphological types of PV+ interneurons all have similar expression of cell adhesion molecules, at least after postnatal day 10 (P10); and 3) between P21 and P25 there is a substantial increase in expression of hemoglobin genes in all subtypes of PV+ interneurons.

Overall, I believe this study is well-executed (with some concerns described below) and will be of broad interest. The idea of discrete transcriptomic cell types versus continua of transcriptomic variability within broad classes is an emerging theme in the field that I think will ultimately lead to a paradigm shift in how we think about and define cell types. Although the novelty is somewhat overstated in places, this study nonetheless represents an important piece of the puzzle in trying to understand cellular diversity in the brain.

Major concerns/suggestions:

1)The anatomic location being studied should be stated in the abstract at a minimum, and also in the title if possible.

2)The significance of the hemoglobin finding would be strengthened if the authors could demonstrate that PV+ interneurons in the cortex also express hemoglobin. This should be a relatively straightforward experiment to do using the same techniques with only a different slicing protocol and comparing PV to SST or some other cell type in adult cortex. The timeline of gene expression onset may be different in a different brain area and is not critical in my opinion to do this across multiple developmental timepoints, just to show that the specificity to PV+ interneurons is generalizable.

3)There are multiple references to transcriptomic and other types of "homogeneity", but this is not precisely defined, and PV cells are not convincingly shown to be more "homogeneous" than other broad cell classes. I think what the authors mean is that the variability in transcriptomic (or other) space does not form discrete sub-clusters, or does not correlate with other modalities. But that doesn't necessarily mean it is uniform. I think these sections should be re-worded slightly to be more clear about what exactly is being shown and how it is quantified.

4)The methods for assigning cells to a morphologic class are not described. I assume this is by expert/manual labeling?

4a)Only cells that could be morphologically characterized as stereotypical AAC, vBC, hBC, vBiC or hBiC were analyzed, and it seems that a large number (37) were excluded even if the morphologic reconstruction was of high quality. This seems somewhat arbitrary, and it would be nice to have a supplementary figure showing some representative morphologies and/or mapping to transcriptomic clusters from this group, and more discussion of the variability here (are these intermediate cell types, or low abundance unique cell types?).

5)The claim that correlation between transcriptome and morphology has not been previously investigated is a straw-man hypothesis and the novelty of this approach is overstated. Several recent papers have also done Patch-seq with detailed morphological reconstructions of the same sequenced cells (see methodology described in Cadwell et al., 2017 and applied in Scala et al., 2019, Scala et al., 2020, and Gouwens et al., 2020). These last three show a similar lack of correspondence between morphology and transcriptomic cell types in cortical interneurons

including PV+ interneurons (Scala et al., 2020 and Gouwens et al., 2020).

6) Given that the authors convincingly show that the PV+ interneurons in CA1 are already fully mature morphologically by P10, and gene expression is only assessed after P10, the claim that cell adhesion molecules do not establish the initial axodendritic structure (and thus the cells' connectivity patterns) is overstated in multiple places but especially in the Results and Discussion sections. It remains entirely possible that CAMs control this process, but it all happens before P10. There is a small acknowledgement of this limitation in the discussion, but only after some bold statements are made.

7) It is unclear whether corrections for multiple comparisons were performed for Figures 3A and 6E (they should be, and this should be stated clearly in the text and/or figure legends).

8) It would be nice to expand the discussion a bit more around the potential significance of hemoglobin expression in PV+ interneurons. There is a suggestion in the final sentence of its possible implications for neuropsychiatric/neurodevelopmental disorders but this could be expanded in the preceding section with a bit more detail about what is currently known or hypothesized for the role of hemoglobin in neurons, and for PV+ neurons in neuropsychiatric disease to make this link at least tenable.

Minor concerns/suggestions:

Abstract:

- Please remove the phrases "surprisingly" and "unexpected" from the abstract, and any other value judgments throughout the text. Better to present the evidence and let the reader interpret for him/herself.
- "diversity created by" – the diversity is not really "created by" the number of cell types but is more reflected in the number of cell types. Different word choice recommended.
- "novel genes whose expression separately identify morphological types but corroborate an overall transcriptomic homogeneity among PV-INS" – this sentence is confusing and somewhat contradictory.
- It would be helpful to have some sort of conclusion/significance statement at the end of the abstract.

Introduction

- "employed single-cell RNA-seq to characterize physiological features" – confusing
- Rather than dedicating an entire paragraph to each of the three hypotheses, it would be easier to follow if all of the necessary background were presented first, and then the three hypotheses as single sentences/phrases in my opinion.
- "or a consequence of neuronal activity" – should be "or as a consequence of neuronal activity"?
- "In the hippocampus, changes during circuit maturation, the development of GABAergic inhibition shows dynamic (Banks et al., 2002; Yu et al., 2006; Fazzari et al., 2010; Salesse et al., 2011) and parvalbumin protein levels continue to increase (Wu et al., 2014)." This is a confusing sentence, is something missing after the word "dynamic"?
- "extensive synaptic regulations have been described to take place in the first 4 postnatal weeks (Luhmann and Prince, 1991)," – it's not clear what is meant by "regulations".
- "morphology-based gene selection" should be explained a bit more. I could understand what you meant after reading the rest of the paper, but when reading the introduction initially it did not make sense.

Results:

- "We hypothesize, but did not test further, that this subunit which underlies the M current (Wang et al., 1998) contributed to the observed physiological changes." Would either remove this unsubstantiated hypothesis or rephrase to make it less strong, i.e. "This subunit has been shown to underlie the M current (Wang et al., 1998) and could potentially contribute to the observed physiological changes." This sentence may also be more appropriate in the Discussion rather than

the Results section.

Discussion:

- "Transcriptomic definition of morphological PV types" – may be better thought of and discussed as transcriptomic correlates given that no gene perfectly "defines" a morphological subtype.

Figures/Data Analysis

- In the figure legend for Figure 2C, this value is referred to as "mapping efficacy" but the Y-axis is simply % of cells. Is this supposed to be a measure of uncertainty of the mapping, or just the number of cells mapped to each class using difference gene selection methods. I could not find a definition of "mapping efficacy" in the Methods section.

- Significant values in Figure 3A should be shown (they are only mentioned in the text).

- In Figure 6E the p-values for the first two plots on the left are identical, would recommend double checking that this is not a mistake.

- All code should be made publicly available upon publication via a repository such as Github to facilitate reproducibility.

References:

CADWELL, C. R., SCALA, F., LI, S., LIVRIZZI, G., SHEN, S., SANDBERG, R., JIANG, X. & TOLIAS, A. S. 2017. Multimodal profiling of single-cell morphology, electrophysiology, and gene expression using Patch-seq. *Nat Protoc*, 12, 2531-2553.

GOUWENS, N. W., SORENSEN, S. A., BAFTIZADEH, F., BUDZILLO, A., LEE, B. R., JARSKY, T., ALFILER, L., ARKHIPOV, A., BAKER, K., BARKAN, E., ET AL. 2020. Toward an integrated classification of neuronal cell types: morphoelectric and transcriptomic characterization of individual GABAergic cortical neurons. *bioRxiv*, 2020.02.03.932244; doi: <https://doi.org/10.1101/2020.02.03.932244>

SCALA, F., KOBAK, D., SHAN, S., BERNAERTS, Y., LATURNUS, S., CADWELL, C.R., HARTMANIS, L., FROUDARAKIS, E., CASTRO, J.R., TAN, Z.H., ET AL. 2019. Layer 4 of mouse neocortex differs in cell types and circuit organization between sensory areas. *Nat Commun*, 10, 4174.

SCALA, F., KOBAK, D., BERNABUCCI, M., BERNAERTS, Y., CADWELL, C. R., CASTRO, J. R., HARTMANIS, L., JIANG, X., LATURNUS, S., MIRANDA, E., ET AL. 2020. Phenotypic variation within and across transcriptomic cell types in mouse motor cortex. *bioRxiv*, 2020.02.03.929158; doi: <https://doi.org/10.1101/2020.02.03.929158>.

Reviewer #2:

Remarks to the Author:

The Authors focus on parvalbumin interneurons and set out to study the relationship of morphology vs. transcriptome complexity. The paper is biased in a number of ways. Firstly, its premise that "the relationship between transcriptomic information and morphology has not been investigated" is incorrect since both the Cadwell et al. and Fuzik et al. (*Nat Biotech*) papers have done exactly so, the latter particularly on CCK interneurons. Secondly, the "morphology guided gene selection" approach takes away the strength of unbiased transcriptomics, bringing lots of technical questions afore. These are summarized below. The authors need to perform benchmarking, technical replicates, better analysis and go for uncharted questions in a revision to make this paper appealing to a broad readership.

1. Technical details were insufficiently described. The whole study seems to be on 75 cells only, which is clearly insufficient. Did these 75 cells also include the ones from the younger animals? (how many per age?) How was gene discovery rate? FDRs? Benchmarking against the Cadwell et al and Fuzik et al studies? How was the %RNA discovery estimated per cell? How do the cells fit

the Zeisel et al. study used in the introduction (again for benchmarking)? How did the authors keep RNA in solution?

2. Instead of SST cells, the authors should minimally use pyramidal cells as controls (to show that there is no duplicate cell or other contamination through identification of Vglut1/2 and using exclusion criteria instead of using Gad1 as "positive control"/inclusion criteria). 127 genes were "enriched" out of how many? How was sequencing performed, i.e. did the authors use Smart-Seq throughout? If so, they have to present read depth etc. to show that RNA retention was appropriate (knowing that fall-outs are numerous with "Patch-seq").

3. It is unlikely that 75 cells with a thresholded approach ("enriched" genes) would cluster at all. This is a false hypothesis. The Authors should use unfiltered data to show if cells cluster. They can take the Fuzik et al. paper as reference, which has exactly done that for CCK interneurons. Likewise, the Authors should use a similar matrix approach as Fuzik et al. did to resolve e.g. molecular vs. biophysical characteristics of PV cells.

4. What do the authors mean with e.g. "morphology associated genes (88)"? My problem here is that they always fall back onto supervised approaches, which is just not rigorous enough. Likewise, the "CAM" study is a turn off – the Authors should focus on what is different and not what is homogeneous. In other terms, the study seems to show that there is no relationship between molecular make-up and morphology/function, which by and large is not true for any other cell type tested so far (by e.g. Patch-seq). Why here then (or does the Harris et al "continuum" really work against this small dataset)? What about minimally reproducing the gap junction maturation parameters that e.g. Marlene Bartos has studied earlier and correlate those with enrichment in presynaptic proteins and receptors/channels?

5. The expression of hemoglobin genes is not new in itself, has been shown for many brain regions earlier. What would be more interesting is addressing functional importance. Is Hg expression an artifact, i.e. no protein being produced and RNA degraded right away? They Authors might touch on one of the critical draw-backs of scRNA-seq, and should prove their point beyond doubt by eg. FACS PV cells in their 100s and run Western blot or MS analysis for target protein discovery.

6. The paper in general is full of technical details. The Authors should weed out their computational bits-and-bobs to make this an enjoyable and focused read. Those details can end up in the SI.

Reviewer #3:

Remarks to the Author:

In the manuscript entitled "Transcriptomic homogeneity and an age-dependent onset of hemoglobin expression characterize morphological PV types" the authors Que et al. generated an impressive dataset consisting of 54 PV neurons with combined patch-clamp recordings, morphological reconstructions and scRNA-seq. An unsupervised analysis of single-cell transcriptomes did not allow for a grouping (or clustering) of PV neurons into morphological types; and morphological types did not match with electrophysiological properties. Nevertheless, a bulk differential gene expression analysis of neurons that were grouped according to morphological criteria resulted in "a hand full" of selective marker genes that passed statistical thresholds. Directing the analysis with 88 such differentially expressed genes allowed resolving finer distinctions also in a single cell dataset. It is not clear how the latter results in a better understanding of PV types. Is the main conclusion from the latter that the ability of PV types to maintain different morphologies relies on the small number of genes that are differentially expressed between each of these types? Or was the intend of this part to make a statement regarding limitations of unsupervised scRNA-seq experiments?

In the second part of the study, the authors focus on the morphological PV type vBC to examine

gene expression dynamics, morphological features and electrophysiological properties between P10 and P77. While no major electrophysiological and morphological change was detected, the authors found an up-regulation of several haemoglobin subunit-coding genes between P21-P25.

The manuscript presents a multimodal single cell RNAseq data set and a comprehensive analysis thereof, which provides potentially interesting insights into the function of hippocampal PV neurons. The data is overall of high quality. However, it is unclear how much new biological insight is provided by the manuscript. The identification of genes per se does not give definitive answers. In summary, I think the manuscript is well suitable for publication in Nature Communications.

Comments:

The term "transcriptomic homogeneity" (e.g. in the title) is misleading. In this paper the authors find differentially expressed genes and proMMT identified different clusters (Fig. S2). So, there is transcriptomic heterogeneity, despite the observation that the 2D visualisation does "not clearly" separate clusters and that there is relatively little heterogeneity compared to the heterogeneity that exists between PV and SST neurons.

What is the main source of heterogeneity in the PV dataset alone (without SST neurons and without mapping to the CA1-IN dataset)? How does a single-cell heat-map look like for the top 10 or 20 marker genes of the four proMMT clusters? Do the top marker genes indicate what the main source of heterogeneity could be functionally related to?

Introduction: What is known about the function of various morphological types?

Figures 7a and S7a do not only show an up regulation of haemoglobin subunit-coding genes between P21-P25, but also a pretty larger number of genes that are down-regulated around that time. What genes are they? A table of all dynamic genes would be useful. Are for some of these genes in situ available on public databases that can confirm the temporal expression dynamics?

In the main text related to Fig. 4F the mapping to "islands" and "bridges to islands" is a bit vague. I would suggest using a cluster algorithm to group cells.

Reviewers' comments:

Reviewer #1 (Remarks to the Author):

This study applied patch-seq to study the correlation between morphology, electrophysiology and gene expression in parvalbumin-positive (PV+) interneurons in CA1 of the hippocampus. The major findings are that 1) there does not seem to be a very good correlation between morphologic subtypes of PV+ neurons, transcriptomic cell types, and electrophysiological properties; 2) despite their different morphologies, different morphological types of PV+ interneurons all have similar expression of cell adhesion molecules, at least after postnatal day 10 (P10); and 3) between P21 and P25 there is a substantial increase in expression of hemoglobin genes in all subtypes of PV+ interneurons.

Overall, I believe this study is well-executed (with some concerns described below) and will be of broad interest. The idea of discrete transcriptomic cell types versus continua of transcriptomic variability within broad classes is an emerging theme in the field that I think will ultimately lead to a paradigm shift in how we think about and define cell types. Although the novelty is somewhat overstated in places, this study nonetheless represents an important piece of the puzzle in trying to understand cellular diversity in the brain.

We thank this Reviewer for his/her comments and detailed suggestions, and appreciate his/her recognition of the importance of this and related studies.

Major concerns/suggestions:

1)The anatomic location being studied should be stated in the abstract at a minimum, and also in the title if possible.

Done.

2)The significance of the hemoglobin finding would be strengthened if the authors could demonstrate that PV+ interneurons in the cortex also express hemoglobin. This should be a relatively straightforward experiment to do using the same techniques with only a different slicing protocol and comparing PV to SST or some other cell type in adult cortex. The timeline of gene expression onset may be different in a different brain area and is not critical in my opinion to do this across multiple developmental timepoints, just to show that the specificity to PV+ interneurons is generalizable.

This is a great suggestion. In order to address this point, we collected six >P25 cortical PV cells from the same transgenic line and processed these together with other hippocampal samples which were also collected during revision. We found that cortical PV cells lack Hb expression. We now show in revised Fig. 7F. To gain further insights into Hb expression patterns in multiple different other cell types, we also analyzed publicly available single-cell RNAseq data, including Cadwell et al. (2016), Gouwens et al. (2020), Tasic et al. (2018), Munoz-Manchado et al. (2018), Fuzik et al. (2016), Zeisel et al. (2015), and Harris et al. (2018) for Hb expression.

Of these, Cadwell's data revealed high and regular expression of Hb subunits in Layer I cortical neurons (non-PV neurogliaform and single bouquet cells). By contrast, other data sets revealed that although Hb subunits can be reliably detected in a subset of single neurons, Hb subunits were absent from most cells. These neurons did not appear to belong to specific types (as defined by the original studies), but represented multiple different cell types.

Of particular importance to our study, we also analyzed hippocampal PV cells in Harris' data set. This analysis, however, did not revealed Hb expression in PV cells. While we can not currently explain this discrepancy, we already point out in Figs. 4 and S7 that the efficacy of gene detection is lower in Harris' data compared to ours and several genes that we detected in our data were not available in their data set. Hb-s potentially fall into this category.

In the revised manuscript, we present this extended analysis on Hb expression in Figs. 7F, S10, and S11.

3)There are multiple references to transcriptomic and other types of “homogeneity”, but this is not precisely defined, and PV cells are not convincingly shown to be more “homogeneous” than other broad cell classes . I think what the authors mean is that the variability in transcriptomic (or other) space does not form discrete sub-clusters, or does not correlate with other modalities. But that doesn't necessarily mean it is uniform. I think these sections should be re-worded slightly to be more clear about what exactly is being shown and how it is quantified.

We thank this reviewer for pointing this out. We revised the text and title to clarify this terminology and now refer to transcriptomic homogeneity (which was our previous phrasing when comparing different morphological PV types) as transcriptomic continuity.

4)The methods for assigning cells to a morphologic class are not described. I assume this is by expert/manual labeling?

Correct. Morphological classification was done by expert/manual labeling. For the revisions, we reconstructed more cells and now display these cell's morphology in the new Fig. 1 and S1.

4a)Only cells that could be morphologically characterized as stereotypical AAC, vBC, hBC, vBiC or hBiC were analyzed, and it seems that a large number (37) were excluded even if the morphologic reconstruction was of high quality. This seems somewhat arbitrary, and it would be nice to have a supplementary figure showing some representative morphologies and/or mapping to transcriptomic clusters from this group, and more discussion of the variability here (are these intermediate cell types, or low abundance unique cell types?).

Thank you for raising this important point. Looking into this issue drew our attention to the fact that the number of cells with ambiguous morphology was erroneously stated in Methods. The correct number was 11 instead of 37.

To address this and the previous point, we morphologically reconstructed more cells. These reconstructions provided us with more clarity regarding their classification, and we have now been able to assign most of these cells to either of the morphological types. Since there was also ambiguity regarding the classification of some of the additionally collected cells, the total number of unclassified cells is now 14 out of 123. We show example uncharacterized cells in revised Fig. S1. Reasons for ambiguity included insufficient axonal recovery to distinguish bistratified from basket cells, some cells had apparent trilaminar axonal morphology, and one cell had both vertical and horizontal dendrites. Although we do not show in separate figure, all these cells mapped together with other PV cells.

5)The claim that correlation between transcriptome and morphology has not been previously investigated is a straw-man hypothesis and the novelty of this approach is overstated. Several recent papers have also done Patch-seq with detailed morphological reconstructions of the same sequenced cells (see methodology described in Cadwell et al., 2017 and applied in Scala et al., 2019, Scala et al., 2020, and Gouwens et al., 2020). These last three show a similar lack of correspondence

between morphology and transcriptomic cell types in cortical interneurons including PV+ interneurons (Scala et al., 2020 and Gouwens et al., 2020).

We thank this Reviewer for this comment. While any relevant conclusion in Cadwell et al. (2017) and Scala et al. (2019) may have been overlooked in our original manuscript, we could not yet include the preprints Scala et al. (2020) and Gouwens et al. (2020) as these very relevant studies became available only after submitting our manuscript. We have now gladly included these and discussed their relevance. As the Reviewer notes, these independent studies also find a lack of correspondence between morphological and transcriptomic PV types, which underscores the finding of our study.

6) Given that the authors convincingly show that the PV+ interneurons in CA1 are already fully mature morphologically by P10, and gene expression is only assessed after P10, the claim that cell adhesion molecules do not establish the initial axondendritic structure (and thus the cells' connectivity patterns) is overstated in multiple places but especially in the Results and Discussion sections. It remains entirely possible that CAMs control this process, but it all happens before P10. There is a small acknowledgement of this limitation in the discussion, but only after some bold statements are made.

We fully agree with this reviewer and did not intend to downplay any role of CAMs in setting up connectivity and axon-dendritic features. Our goal is to simply state the observation that CAM expression between mature (P>21) PV types is not that different. We revised the manuscript to make this point very clear.

7) It is unclear whether corrections for multiple comparisons were performed for Figures 3A and 6E (they should be, and this should be stated clearly in the text and/or figure legends).

These were corrected for multiple comparisons. We now clarify this by explicitly referring to these statistical measures as “FDR” throughout the text.

8) It would be nice to expand the discussion a bit more around the potential significance of hemoglobin expression in PV+ interneurons. There is a suggestion in the final sentence of its possible implications for neuropsychiatric/neurodevelopmental disorders but this could be expanded in the preceding section with a bit more detail about what is currently known or hypothesized for the role of hemoglobin in neurons, and for PV+ neurons in neuropsychiatric disease to make this link at least tenable.

We appreciate this Reviewer's comment. Our data reveals expression of hemoglobin subunits in PV cells, but does not provide further insights as to whether these transcripts are translated and functionally assembled. Added to this, as we already mentioned in the discussion, hemoglobins are implicated in multiple different functions, making their role in neurons controversial. Since our data does not clarify this issue, we chose to scale down on the disease relevance instead of further expanding.

Minor concerns/suggestions:

We thank this Reviewer for the following comments and suggestions. We revised the manuscript accordingly.

Abstract:

- Please remove the phrases “surprisingly” and “unexpected” from the abstract, and any other value judgments throughout the text. Better to present the evidence and let the reader interpret for him/

herself. - **Revised.**

- “diversity created by” – the diversity is not really “created by” the number of cell types but is more reflected in the number of cell types. Different word choice recommended. - **Revised.**

- “novel genes whose expression separately identify morphological types but corroborate an overall transcriptomic homogeneity among PV-INS” – this sentence is confusing and somewhat contradictory. - **Revised.**

- It would be helpful to have some sort of conclusion/significance statement at the end of the abstract. - **Included.**

Introduction

- “employed single-cell RNA-seq to characterize physiological features” – confusing - **Revised.**

- Rather than dedicating an entire paragraph to each of the three hypotheses, it would be easier to follow if all of the necessary background were presented first, and then the three hypotheses as single sentences/phrases in my opinion. - **We revised the introduction according to this suggestion.**

- “or a consequence of neuronal activity” – should be “or as a consequence of neuronal activity”? - **Revised.**

- “In the hippocampus, changes during circuit maturation, the development of GABAergic inhibition shows dynamic (Banks et al., 2002; Yu et al., 2006; Fazzari et al., 2010; Salesse et al., 2011) and parvalbumin protein levels continue to increase (Wu et al., 2014).” This is a confusing sentence, is something missing after the word “dynamic”? - **Revised.**

- “extensive synaptic regulations have been described to take place in the first 4 postnatal weeks (Luhmann and Prince, 1991),” – it’s not clear what is meant by “regulations”. - **Revised.**

- “morphology-based gene selection” should be explained a bit more. I could understand what you meant after reading the rest of the paper, but when reading the introduction initially it did not make sense. - **Revised.**

Results:

- “We hypothesize, but did not test further, that this subunit which underlies the M current (Wang et al., 1998) contributed to the observed physiological changes.” Would either remove this unsubstantiated hypothesis or rephrase to make it less strong, i.e. “This subunit has been shown to underly the M current (Wang et al., 1998) and could potentially contribute to the observed physiological changes.” This sentence may also be more appropriate in the Discussion rather than the Results section. - **Removed.**

Discussion:

- “Transcriptomic definition of morphological PV types” – may be better thought of and discussed as transcriptomic correlates given that no gene perfectly “defines” a morphological subtype. - **Revised.**

Figures/Data Analysis

- In the figure legend for Figure 2C, this value is referred to as “mapping efficacy” but the Y-axis is simply % of cells. Is this supposed to be a measure of uncertainty of the mapping, or just the number of cells mapped to each class using difference gene selection methods. I could not find a

definition of “mapping efficacy” in the Methods section. - **We defined mapping efficacy as % of cells mapped to a category. We renamed this “Fraction of cells mapped (%)” to make this definition clearer.**

- Significant values in Figure 3A should be shown (they are only mentioned in the text). - **Done.**

- In Figure 6E the p-values for the first two plots on the left are identical, would recommend double checking that this is not a mistake. - **We checked these values: p-values are different, but p-adjusted is the same. Our code uses multiple tests from statsmodels (<https://www.statsmodels.org/dev/generated/statsmodels.stats.multitest.multipletests.html>) with the Benjamini-Hochberg Procedure.**

- All code should be made publicly available upon publication via a repository such as Github to facilitate reproducibility. - **Our codes will be available:** <https://github.com/foldy-lab/Transcriptomics-of-Pvalb-Cells>

References:

CADWELL, C. R., SCALA, F., LI, S., LIVRIZZI, G., SHEN, S., SANDBERG, R., JIANG, X. & TOLIAS, A. S. 2017. Multimodal profiling of single-cell morphology, electrophysiology, and gene expression using Patch-seq. *Nat Protoc*, 12, 2531-2553.

GOUWENS, N. W., SORENSEN, S. A., BAFTIZADEH, F., BUDZILLO, A., LEE, B. R., JARSKY, T., ALFILER, L., ARKHIPOV, A., BAKER, K., BARKAN, E., ET AL. 2020. Toward an integrated classification of neuronal cell types: morphoelectric and transcriptomic characterization of individual GABAergic cortical neurons. *bioRxiv*, 2020.02.03.932244; doi: <https://doi.org/10.1101/2020.02.03.932244>

SCALA, F., KOBAK, D., SHAN, S., BERNAERTS, Y., LATURNUS, S., CADWELL, C.R., HARTMANIS, L., FROUDARAKIS, E., CASTRO, J.R., TAN, Z.H., ET AL. 2019. Layer 4 of mouse neocortex differs in cell types and circuit organization between sensory areas. *Nat Commun*, 10, 4174.

SCALA, F., KOBAK, D., BERNABUCCI, M., BERNAERTS, Y., CADWELL, C. R., CASTRO, J. R., HARTMANIS, L., JIANG, X., LATURNUS, S., MIRANDA, E., ET AL. 2020. Phenotypic variation within and across transcriptomic cell types in mouse motor cortex. *bioRxiv*, 2020.02.03.929158; doi: <https://doi.org/10.1101/2020.02.03.929158>.

Reviewer #2 (Remarks to the Author):

The Authors focus on parvalbumin interneurons and set out to study the relationship of morphology vs. transcriptome complexity. The paper is biased in a number of ways. Firstly, its premise that “the relationship between transcriptomic information and morphology has not been investigated” is incorrect since both the Cadwell et al. and Fuzik et al. (*Nat Biotech*) papers have done exactly so, the latter particularly on CCK interneurons. Secondly, the “morphology guided gene selection” approach takes away the strength of unbiased transcriptomics, bringing lots of technical questions afore. These are summarized below. The authors need to perform benchmarking, technical replicates, better analysis and go for uncharted questions in a revision to make this paper appealing to a broad readership.

We thank this reviewer for his/her critical comments. The relationship between transcriptomic information and morphology is not yet understood. Other similar preprints (Gouwens et al., 2020; Scala et al., 2020, both in cortex) also aim to solve this problem, and draw similar conclusions to ours, and we therefore believe the questions outlined in this study are currently important issues.

1. Technical details were insufficiently described. The whole study seems to be on 75 cells only, which is clearly insufficient. Did these 75 cells also include the ones from the younger animals? (how many per age?) How was gene discovery rate? FDRs? Benchmarking against the Cadwell et al and Fuzik et al studies? How was the %RNA discovery estimated per cell? How do the cells fit the Zeisel et al. study used in the introduction (again for benchmarking)? How did the authors keep RNA in solution?

In order make our study more complete, we collected and included additional morphologically classified 13 cells in the revised manuscript. We now have a total of 88, ranging from p10-77 morphologically classified and sequenced PV cells. The number of cells and their break-down into age-groups are indicated in the text and/or figures.

Using the bioinformatic pipeline used in our paper, we benchmarked our data against Zeisel et al. (2015), Cadwell et al. (2016), Fuzik et al. (2016), and other single cell data (see below and new Fig. S2):

Based on these numbers, we believe that our study has comparably high quality when benchmarked against these other existing studies.

Regarding the question of RNA retention, we used a protocol that is described in section 'Sample collection' of Methods.

2. Instead of SST cells, the authors should minimally use pyramidal cells as controls (to show that there is no duplicate cell or other contamination through identification of Vglut1/2 and using exclusion criteria instead of using Gad1 as "positive control"/inclusion criteria). 127 genes were "enriched" out of how many? How was sequencing performed, i.e. did the authors use Smart-Seq throughout? If so, they have to present read depth etc. to show that RNA retention was appropriate (knowing that fall-outs are numerous with "Patch-seq").

- Regarding pyramidal cell as controls: we now include a more complete set of marker genes (GABAergic, MGE, CGE, glutamatergic, including Vglut1 and Vglut2) in Fig. S2. We also would like to emphasize that Gad1 expression (or any marker shown in Fig 1) was not part of the inclusion criteria.
- Regarding total gene numbers: we now indicate the total number of genes (N=35,825, using Ensembl's GRCm38.v95 reference genome) in the main text.
- Regarding sequencing: as described in Methods, all sequencing was performed with Clontech's SMARTer Ultra Low RNA Input v4 or SMART-Seq HT kit.
- Regarding presumed fall out rates: under the assumption that all cell types express approximately a similar number of unique genes and isoforms, % RNA discovery and gene discovery rate are both linearly proportional to the number of unique isoforms and number of unique genes detected, respectively. As such, as indicated by the benchmarking above, our % RNA and gene discovery rates is comparable to other datasets. For example, our study had ~2x as many unique isoforms and ~1.5x as many unique genes as the Zeisel et al. study, therefore our % RNA discovery rate and our gene discovery rates are ~2x and ~1.5x of the respective rates in that study. This also suggests that fall out rates are not elevated with patch-seq; indeed, as this Reviewer implies, fall out rate is proportional to read depth, which does not necessarily depend on the method of cell or RNA collection.

3. It is unlikely that 75 cells with a thresholded approach ("enriched" genes) would cluster at all. This is a false hypothesis. The Authors should use unfiltered data to show if cells cluster. They can take the Fuzik et al. paper as reference, which has exactly done that for CCK interneurons. Likewise, the Authors should use a similar matrix approach as Fuzik et al. did to resolve e.g. molecular vs. biophysical characteristics of PV cells.

We thank this reviewer for raising these important points.

- Regarding the use of unfiltered data: in Fig 1D (where morphological information is displayed but not used for clustering), we already show that our PV cells did not separately cluster, whether or not SST cells were included. To make the analysis shown in Fig. 1D even less filtered, here we show a UMAP of cells without implementing any prior feature selection or gene enrichment. This still shows lack of any clustering among PV-INs, similarly to Fig 1D (color codes are same as in main figure).

- Regarding Fuzik's approach: Thank you for suggesting the use of this approach. We implemented it and the results are in agreement with cell distributions found in the proMNT types. We shown these in revised Fig. 2B.

- Regarding the number of cells used for clustering: we tested if restricting our analysis to <75 cells would limit our ability to detect clusters, granted that cells were transcriptomically different. For this test we used two different data sets from previous studies. (1) SST-OLM interneurons (n=23, Winterer et al., 2019) and subicular burst-spiking (BS, n=21) and regular

firing (RS, n=13) pyramidal cells (Földy et al., 2016). We choose these types because SST-OLM cells are expected to be different from BS and RS cells, whereas the two pyramidal cells were found to be transcriptomically similar in the original study. (2) Sst.Npy (n=386), Pvalb.C1ql1 (n=211) and Cck.Cxcl14 (n=465) cells, which are all different from each other, from Harris et al. (2018). Data set (1) was generated using patch-seq, whereas data set (2) was generated after FACS.

Using the same methods as in the paper and randomly selecting a subset of cells from these data, we evaluated clustering at low sample numbers, i.e. at cell numbers 15, 30 and 45.

Using data set (1), we found separate clustering of SST-OLMs with only 15 cells, whereas the two transcriptomically similar pyramidal types did not separately cluster even when the complete data set was used. Using data set (2), we already see separate clustering of the three types with 30 cells, even though these data have much lower gene counts (see benchmarking above).

This demonstrates that our methods are capable of capturing separate clusters at low sample numbers (<30) if cell types sufficiently differ from each other, and furthermore recapitulates a scenario that is similar to what we show in Fig 1. for PV-INs. We now include this demonstration in new Fig. S3.

4. What do the authors mean with e.g. "morphology associated genes (88)"? My problem here is that they always fall back onto supervised approaches, which is just not rigorous enough. Likewise, the "CAM" study is a turn off – the Authors should focus on what is different and not what is homogeneous. In other terms, the study seems to show that there is no relationship between molecular make-up and morphology/function, which by and large is not true for any other cell type tested so far (by e.g. Patch-seq). Why here then (or does the Harris et al "continuum" really work against this small dataset)? What about minimally reproducing the gap junction maturation parameters that e.g. Marlene Bartos has studied earlier and correlate those with enrichment in presynaptic proteins and receptors/channels?

As to the reviewer's comment on the rigor of supervised approaches, it is generally easier to identify differences within a dataset using supervised methods. However, this potentially causes issues of reproducibility when there isn't a separate test set to validate the results on. As such, the reviewer is correct that a supervised method on a limited dataset is suboptimal for demonstrating the existence of differences. However, when it comes to demonstrating a lack of separation, the failure of a supervised method to find separation provides stronger

evidence than the failure of an unsupervised method, as it is inherently more sensitive to any such differences between a priori labeled categories. As we found similarity between morphological PV types in our paper, we feel that a more rigorous, supervised approach would be superior to an unsupervised one.

As other recent patch-seq preprints attest, the community is interested in this supervised approach for the very purpose of addressing the key problem of whether there is a correspondence between morphology and transcriptomic content (Scala et al., 2020, bioRxiv; Gouwens et al., 2020, bioRxiv). The findings of Scala and Gouwens, as Reviewer 1 also highlights, “show a similar lack of correspondence between morphology and transcriptomic cell types in cortical interneurons including PV+ interneurons (Scala et al., 2020 and Gouwens et al., 2020).” However, as this Reviewer also points it out, this property may be more pronounced in PV-INs than in other cell types, as for example in cortical SST cells there is a larger correspondence between transcriptomic and other modalities (Tasic et al., 2018; Nigro et al., 2018; Naka et al., 2019).

With the CAM analysis, we tested a key hypothesis related to cellular connectivity and a homogenous CAM expression is the outcome. We believe that there is an audience for this and therefore prefer to keep this in the manuscript.

5. The expression of hemoglobin genes is not new in itself, has been shown for many brain regions earlier. What would be more interesting is addressing functional importance. Is Hg expression an artifact, i.e. no protein being produced and RNA degraded right away? They Authors might touch on one of the critical draw-backs of scRNA-seq, and should prove their point beyond doubt by eg. FACS PV cells in their 100s and run Western blot or MS analysis for target protein discovery.

We fully agree with this Reviewer that elucidating any Hb function in neurons, and in PV-INs in particular, would be important. As highlighted in Discussion, this question remains open. However, given the resources and time it would take, we feel that solving this problem is beyond the scope of this current study and would instead motivate a follow up study.

Even if Hb-s are not translated in PV-INs, their expression onset remains an important observation. In this case, Hb-s will add to a set of genes which are expressed but not translated in PV-INs. For example, cholecystokinin and Camk2a, which peptide/protein are considered to be cell type specific markers for neighboring CCK-INs and pyramidal cells, respectively, are also expressed as mRNA in PV-INs.

6. The paper in general is full of technical details. The Authors should weed out their computational bits-and-bobs to make this an enjoyable and focused read. Those details can end up in the SI.

In revising this paper, we put any further technical details in the SI to keep the manuscript as focused as possible.

Reviewer #3 (Remarks to the Author):

In the manuscript entitled “Transcriptomic homogeneity and an age-dependent onset of hemoglobin expression characterize morphological PV types” the authors Que et al. generated an impressive dataset consisting of 54 PV neurons with combined patch-clamp recordings, morphological reconstructions and scRNA-seq. An unsupervised analysis of single-cell transcriptomes did not allow for a grouping (or clustering) of PV neurons into morphological types; and morphological types did not match with electrophysiological properties. Nevertheless, a bulk differential gene expression analysis of neurons that were grouped according to morphological criteria resulted in "a hand full" of selective marker genes that passed statistical thresholds. Directing the analysis with 88

such differentially expressed genes allowed resolving finer distinctions also in a single cell dataset. It is not clear how the latter results in a better understanding of PV types. Is the main conclusion from the latter that the ability of PV types to maintain different morphologies relies on the small number of genes that are differentially expressed between each of these types? Or was the intent of this part to make a statement regarding limitations of unsupervised scRNA-seq experiments?

In the second part of the study, the authors focus on the morphological PV type vBC to examine gene expression dynamics, morphological features and electrophysiological properties between P10 and P77. While no major electrophysiological and morphological change was detected, the authors found an up-regulation of several haemoglobin subunit-coding genes between P21-P25.

The manuscript presents a multimodal single cell RNAseq data set and a comprehensive analysis thereof, which provides potentially interesting insights into the function of hippocampal PV neurons. The data is overall of high quality. However, it is unclear how much new biological insight is provided by the manuscript. The identification of genes per se does not give definitive answers. In summary, I think the manuscript is well suitable for publication in Nature Communications.

We thank this Reviewer for carefully reading our manuscript and for his/her comments. We incorporated these comments to improve the manuscript.

Comments:

The term "transcriptomic homogeneity" (e.g. in the title) is misleading. In this paper the authors find differentially expressed genes and proMMT identified different clusters (Fig. S2). So, there is transcriptomic heterogeneity, despite the observation that the 2D visualisation does "not clearly" separate clusters and that there is relatively little heterogeneity compared to the heterogeneity that exists between PV and SST neurons.

We revised the manuscript to clarify this terminology and now we refer to it as transcriptomic continuity.

What is the main source of heterogeneity in the PV dataset alone (without SST neurons and without mapping to the CA1-IN dataset)? How does a single-cell heat-map look like for the top 10 or 20 marker genes of the four proMMT clusters? Do the top marker genes indicate what the main source of heterogeneity could be functionally related to?

Thank you for this great suggestion. We reanalyzed our now extended data set, which also refined the identity of proMMT types, and plotted all (n=25) significantly enriched genes between any comparison of the 4 proMMT types. This revealed that PV cells divide into two main groups and that the *Synpr* proMMT type largely, but not completely, conform to the BIC type, whereas the 3 *Pthlh* types to non-BIC type. The identity and composition of the proMMT types was also confirmed by Fuzik's approach, which was suggested to be used by Reviewer 2. These analyses are now included in revised Fig. 2. Overall, these added analyses revealed an agreement between the unsupervised (Fig. 2) and morphology-based supervised results (Fig. 4), further strengthening our original conclusion.

Introduction: What is known about the function of various morphological types?

Thank you for this suggestion. We added information about the *in vivo* function of these cells to the Introduction.

Figures 7a and S7a do not only show an up regulation of haemoglobin subunit-coding genes between P21-P25, but also a pretty larger number of genes that are down-regulated around that

time. What genes are they? A table of all dynamic genes would be useful. Are for some of these genes in situ available on public databases that can confirm the temporal expression dynamics?

Our extended data set revealed a more refined and smaller set of down regulating genes. All these genes are shown in the figure, discussed in the Discussion, and their expression statistics are included in the Supplementary Excel file. Furthermore, to provide independently generated information about the dynamics of their expression, we now include in situ data from the Allen Mouse Brain Atlas representing their ~2 month old adult brain expression (Fig. S9). For reference, we also include Pvalb expression pattern. As the figure shows, most genes that we detected as down-regulated are not expressed in the adult hippocampus. Or, if expressed, their expression pattern does not appear to overlap with that of Pvalb. Hence, these data confirm our finding regarding the lack / developmental down-regulation of these genes in PV-INS.

In the main text related to Fig. 4F the mapping to “islands” and “bridges to islands” is a bit vague. I would suggest using a cluster algorithm to group cells.

Thank you for this suggestion. We used K-Means clustering to look into this issue. The cells mapping to “islands” 1 and 2 clustered together (not very surprising since there are very few cells in island 2, and they are very close to island 1), but cells mapping to “islands” 3 and 4 clustered separately. In this manner, we identified three separate clusters, which we highlighted in the figure. We now directly refer to these clusters in the text without using the island terminology.

Reviewers' Comments:

Reviewer #1:

Remarks to the Author:

I commend the authors for the substantial amount of work they have put into this revision, and for their thoughtful responses to the reviewer comments. They have adequately addressed all of my concerns and I believe the manuscript is suitable for publication in Nature Communications.

Reviewer #2:

Remarks to the Author:

The Authors have certainly made a laudable effort to update and revise some parts of their study. However, I remain critical of their notions and maintain that many of my original questions were not addressed fully. These are:

1) Subclasses: The authors argue in their response that a supervised approach is more stringent than an unsupervised one. They also argue that there is a "continuum" of PV cells. Yet in their response to Referee #3 they write: "This revealed that PV cells divide into two main groups and that the Synpr proMMT type largely, but not completely, conform to the BIC type, whereas the 3 Pthlh types to non-BIC type." This sounds like a paradox. If candidate genes exist that split PV cells into subtypes then how can one assume "continuity"? Using a candidate approach, one could list out the top 25+ genes that are different and correlate with morphological/electrophysiological criteria and trash the "continuity" idea.

2) Hemoglobin data: I understand that the authors want to be "first" with this sort of analysis, and that papers coming through biorxiv often tend to cause panic. Yet I cannot accept that for a reason to allow the haemoglobin expression data to pass peer-review. As the authors suggest, other cells have many RNAs that are never translated. But then, the haemoglobin data can equally well be meaningless. If so, it is an arbitrary observation that is grossly over-emphasised since it will be only relevant to a small number of investigators who actually run patch-see protocols but none other in the community (because it is physiologically irrelevant). In turn, if protein is expressed too then the situation changes significantly. I also do not understand the argument that what was asked is time consuming. If some histochemistry can be done beyond P10 that will take a week or two. Alternative methods are available, too.

3) Developmental time points: How did the authors control read depth, batch effects for successive time-points? There is a small number of high-profile developmental biology papers whose central question was exactly this and use sophisticated tools to test differentiation trajectories (e.g. RNA-velocity).

4) Morphology x electrophysiology x RNA expression matrices: I wonder why the authors did not use a matrix as suggested. It would illustrate randomness clearly. Likewise, it could help in distinguishing patterns in subgroups. Since prior studies benefited from this approach one might wonder why the authors are reluctant to perform such analysis.

Minor: The title sounds quite strange; "morphological PV types" is an unusual expression. Is there a non-morphological PV cell type? Shall this not be "morphologically-distinct" perhaps?

Reviewer #3:

Remarks to the Author:

The authors have reanalysed and extended their initial dataset, and now provide a more complete picture of the relationship between morphology, electrophysiology and gene expression of PV+

interneurons in CA1 of the hippocampus. All of my comments and criticisms have been addressed adequately. I think the manuscript is well suitable for publication in Nature Communications.

REVIEWER COMMENTS

Reviewer #1 (Remarks to the Author):

I commend the authors for the substantial amount of work they have put into this revision, and for their thoughtful responses to the reviewer comments. They have adequately addressed all of my concerns and I believe the manuscript is suitable for publication in Nature Communications.

We thank this Reviewer for his/her time, effort, and constructive critiques.

Reviewer #2 (Remarks to the Author):

The Authors have certainly made a laudable effort to update and revise some parts of their study. However, I remain critical of their notions and maintain that many of my original questions were not addressed fully. These are:

1) Subclasses: The authors argue in their response that a supervised approach is more stringent than an unsupervised one. They also argue that there is a "continuum" of PV cells. Yet in their response to Referee #3 they write: "This revealed that PV cells divide into two main groups and that the Synpr proMMT type largely, but not completely, conform to the BIC type, whereas the 3 Pthlh types to non-BIC type." This sounds like a paradox. If candidate genes exist that split PV cells into subtypes then how can one assume "continuity"? Using a candidate approach, one could list out the top 25+ genes that are different and correlate with morphological/electrophysiological criteria and trash the "continuity" idea.

Originally, we used the term *homogeneity*. In response to suggestions from R1 & R3, we considered this issue further, and selected the word *continuum* as the quoted passage suggest there is not a very clear separation (hence, „not completely“, see also Fig. 4D) between groups. The manuscript does indeed list the „top“ genes that best separate the groups (see Figure S6) following support vector machine classification— independent of whether these genes are significantly differentially expressed between groups. Even with that approach, we could not find a clear separation between the groups, supporting the idea of a transcriptomic continuum in PV cells.

2) Hemoglobin data: I understand that the authors want to be "first" with this sort of analysis, and that papers coming through biorxiv often tend to cause panic. Yet I cannot accept that for a reason to allow the haemoglobin expression data to pass peer-review. As the authors suggest, other cells have many RNAs that are never translated. But then, the haemoglobin data can equally well be meaningless. If so, it is an arbitrary observation that is grossly over-emphasised since it will be only relevant to a small number of investigators who actually run patch-see protocols but none other in the community (because it is physiologically irrelevant). In turn, if protein is expressed too then the situation changes significantly. I also do not understand the argument that what was asked is time consuming. If some histochemistry can be done beyond P10 that will take a week or two. Alternative methods are available, too.

In our original response, we listed several biorxiv papers with the intention of demonstrating how other recent findings are consistent with our own, and give further confidence in our claims.

With respect to the hemoglobin data: we are aware that protein data would increase the impact of this work and to this end—as suggested by this Reviewer—had tried multiple commercial antibodies in slice prepared from >1 month old animals. However, we could not unambiguously confirm hemoglobin protein expression.

Staining was overall either weak or ubiquitous, a pattern furthermore inconsistent with available in situ data from the Allen Brain Institute (see below figure). Specifically, while alpha subunit antibodies ab92492 (Abcam) and abx129868 (Abexxa) positively labeled PV cells (panel B), they also labeled granule, CA3 and CA1 pyramidal cells, a pattern of which is inconsistent with Allen’s Hba-a1 in situ pattern (panel A). When testing for beta subunit (panels C and D), with one antibody, we found a similar pattern (abx101169, Abbexa) and with another, no expression (sc-390668, Santa Cruz).

Given that Hb is an essential and widely expressed protein, and the consequent challenges in raising Abs against Hbs as well testing their specificity in KO-s, this is not entirely surprising. Since we cannot confirm Hb expression, we have decided to tone down the claims about Hb by specifying that it is mRNA expression; we are also willing to remove Hb from the title altogether, if the editor so requests.

3) Developmental time points: How did the authors control read depth, batch effects for successive time-points? There is a small number of high-profile developmental biology papers whose central question was exactly this and use sophisticated tools to test differentiation trajectories (e.g. RNA-velocity).

As to batch effects in successive sequencing runs, cells collected from differently aged animals were mostly intermixed during successive batches, and no other control was applied. Here, we additionally plot the expression rate of genes belonging to four different mitochondrial and nuclear categories across subsequent sequencing batches, which show consistent behavior. (Expression rate is how many genes detected out of n, the number of genes for the respective GO term.)

In regard of the read depth for successive age time-points, we plotted the number of sequenced (red) and aligned (blue) reads versus the cell age (left panel), revealing that successive time-points do not correlate with read depth. In addition, we did a sub-sampling, where we looked up the cell with the least number of aligned reads that passed quality control and based on this generated a copy of our data in which we randomly sampled the same amount of reads per cell (weighted by the gene distribution in the cell) before running the TPM normalization. Pearson correlation comparison (middle panel) between the actual TPM values and the sub-sampled TPM values (scale from 0-1; correlations calculated on $\text{Log}_2(1+\text{TPM})$, with genes dropped that were expressed in less than 5% of the cells) revealed that the sequencing depth was sufficiently large that sub-sampling all cells to the lowest sequencing depth would have negligible effects on the results. Additional plot (right panel) highlight this further by showing the correlations of cells against their own sub-sampling (blue) and against sub-sampling of other cells (red), revealing a high correlation to themselves and no correlation to others. Using these posthoc analyses, we find it highly unlikely that variation in sequencing depth has adversely affected our results.

Finally, we also explored the possibilities of using cell trajectory programs as suggested by the Reviewer. While our paper already contains the results of using a supervised approach to *Monocle* (Figure S9A and S9B), in response to this suggestion we have run *velocyto* (which has only unsupervised options) to calculate RNA-velocity in our data. We ran the program both on all PV cells (colored by either age or morphological types, left and middle panels, respectively), and on only vBCs (colored by age; right panel). These did not reveal any clear trajectories.

This outcome was not entirely surprising. RNA-velocity, while a powerful concept, was initially developed for two potential cases: for the analysis of cell differentiation during development (and not maturation as our study covers), and in cases where cells were collected from different stages of some cycle that they undergo, such as cell or circadian cycles. As all of our cells were collected beyond developmental cell division, and we did not collect them at all times of day to study the circadian cycle in our paper, neither effects are present in our data.

4) Morphology x electrophysiology x RNA expression matrices: I wonder why the authors did not use a matrix as suggested. It would illustrate randomness clearly. Likewise, it could help in distinguishing patterns in subgroups. Since prior studies benefited from this approach one might wonder why the authors are reluctant to perform such analysis.

Upon previous suggestion of R2, we had included the referred matrix in fig. 2b in our initial resubmission, in which we compare morphologically-defined with the transcriptomically-defined PV cell types. It is important to note that we were not able to conduct similar matrices in relation to electrophysiology, as the dataset has been shown to be homogeneous.

Minor: The title sounds quite strange; "morphological PV types" is an unusual expression. Is there a non-morphological PV cell type? Shall this not be "morphologically-distinct" perhaps?

We thank the reviewer for the suggestion and made adjustments accordingly to the title and manuscript.

Reviewer #3 (Remarks to the Author):

The authors have reanalysed and extended their initial dataset, and now provide a more complete picture of the relationship between morphology, electrophysiology and gene expression of PV+ interneurons in CA1 of the hippocampus. All of my comments and criticisms

have been addressed adequately. I think the manuscript is well suitable for publication in Nature Communications.

We thank this Reviewer for his/her time, effort, and constructive critiques.

Reviewers' Comments:

Reviewer #2:

Remarks to the Author:

The Authors have addressed my concerns.

I have made clear recommendations to help the handling editor, particularly since I think showing negative data is as important as highlighting positive ones. Therefore, all data from the last rebuttal letter (IHC, RNA velocity, matrices) shall be lifted over to the SI of the paper. It will be important to, e.g., demonstrate that despite RNA expression, haemoglobin subunit localisation is anything but convincing in neurons. These amends will place the findings into a realistic order of priority, and will be most educational to the broader scientific community.

Otherwise, I think the paper is ready for publication.

REVIEWERS' COMMENTS

Reviewer #2 (Remarks to the Author):

The Authors have addressed my concerns.

I have made clear recommendations to help the handling editor, particularly since I think showing negative data is as important as highlighting positive ones. Therefore, all data from the last rebuttal letter (IHC, RNA velocity, matrices) shall be lifted over to the SI of the paper. It will be important to, e.g., demonstrate that despite RNA expression, haemoglobin subunit localisation is anything but convincing in neurons. These amends will place the findings into a realistic order of priority, and will be most educational to the broader scientific community.

Otherwise, I think the paper is ready for publication.

We thank this Reviewer for his/her comments. We moved the read depth quality controls into the SI. However, we prefer not to include the other analyses in SI as they are not negative data as the Reviewer suggests, but inconclusive. For example, since we could not confirm the specificity of Hb antibodies, the IHC data neither supports, nor rejects the possibility that translated hemoglobins are present in PV-INs. We believe that this Peer Reviewer File will sufficiently document these efforts for those readers who are interested.